

# Dynamics of salt intrusion in complex estuarine networks; an idealised model applied to the Rhine-Meuse Delta

Bouke Biemond[1], Wouter Kranenburg[2,3], Ymkje Huismans[2,3], Huib E. de Swart[1], and Henk A. Dijkstra[1]

[1]Institute for Marine and Atmospheric research Utrecht, Department of Physics, Utrecht University, Utrecht, the Netherlands
[2]Marine and Coastal Systems, Deltares, Delft, the Netherlands
[3]Faculty of Civil Engineering and Geosciences, Delft University of Technology, Delft, The Netherlands

**Correspondence:** Bouke Biemond (w.t.biemond@uu.nl)

**Abstract.** Many deltas in the world consist of a network of connected channels. We identify and quantify characteristics of salt intrusion in such systems with use of an idealised model. The Rhine-Meuse Delta is selected as a prototype example of a complex network with many channels. The model is able to capture the characteristics of the tide-dominated water level variations due to the main tidal component and the salinity time series for one year of observations. Quantification of tidally averaged salt transport components shows that transport related to exchange flow is dominant in the seaward, deep parts of the network, but tidal dispersion is dominant in shallower channels further inland. Close to the network junctions, a tidally averaged downgradient salt transport is generated by the tidal currents, which is explained from the phase differences between the tidal currents in the different channels. Salt overspill is confined to the most seaward part of the Rhine-Meuse Delta. The magnitudes of the response times of different channels to changes in discharge increases with distance to the estuary mouth, and with decreasing net water transport through the channel. In channels without a subtidal discharge, response times are a factor 2-4 larger than in the other channels. The effect of changes to the depth on the extent of salt intrusion strongly depends on where the change takes place. If the change is within the salt intrusion range, deepening will locally increase salt intrusion due to an increase in salt transport by the exchange flow. If the change is outside the salt intrusion range, changes to the net water transport dominate the response of the salt intrusion.

## 1 Introduction

The extent of salt intrusion in estuaries is changing worldwide, as a result of climate change and anthropogenic activities. Climate change includes e.g. sea level rise (Qiu and Zhu, 2015) and changes in freshwater discharge (Bellafiore et al., 2021). Examples of anthropogenic activities are dam constructions in rivers for water storage and hydropower generation (Qiu and Zhu, 2013), channel deepening for ship navigation and construction of polders (Liu et al., 2019). The availability of theoretical frameworks to explain, and suitable models to simulate salt intrusion, are crucial to predict how these changes affect the extent of salt intrusion.

For single-channel estuaries, relationships between estuarine geometry, forcing conditions and salt intrusion are extensively studied. For this, theoretical frameworks (Geyer and MacCready, 2014), semi-empirical models (Savenije, 1993), idealised



models (MacCready, 2004; Wei et al., 2022), models of intermediate complexity (Dijkstra and Schuttelaars, 2021) and detailed
numerical models (Ralston et al., 2010; Martyr-Koller et al., 2017; Liu et al., 2024) are being used.

However, a number of estuaries consist of multiple channels. For these systems, also referred to as estuarine networks, the
theories developed for single-channel estuaries need to be extended, as they behave differently in several ways. Observations
show (Gong et al., 2014) that a tidally averaged salt transport between channels through the junctions occurs, a phenomenon
known as salt overspill. Second, the phasing of tidal currents in estuarine networks creates local minima in the salt field
(Warner et al., 2002; Garcia et al., 2022), which challenges the traditional view that salt transport by tides can be considered
as a dispersive process (Winterwerp, 1983). The implications of these phenomena for the large-scale salt dynamics is hard to
determine from observations. Furthermore, unsteadiness of the salt field is known to be important to single-channel estuaries
(e.g. Bowen and Geyer (2003); Banas et al. (2004)). For estuarine networks, Biemond et al. (2023) find that the interaction
between channels depends on forcing conditions. Observations show (Eslami et al., 2021) that salt intrusion in some channels
in the Mekong Delta changes substantially due to variation in tidal forcing over the spring-neap cycle, while other channels are
less sensitive. Using a 3D model, they explained this from differences in depth and stratification, but the interaction between the
channels was not explicitly quantified. Next, theory about the effect of changes to the geometry of estuaries does not contain all
the relevant processes in estuarine networks. For example, Wu et al. (2016) suggest that deepening of the North Passage, one
of the channels of the Yangtze River Estuary, may have contributed to salinity of the other channels. Liu et al. (2019) mention
that channel deepening changed the amount of discharge reaching the Modoamen Waterway in the Pearl River Delta, but that
local channel deepening was more important to the salt intrusion.

Salt intrusion in estuarine networks has been studied with one-dimensional (1D) models, e.g. by Nguyen and Savenije (2006)
and Zhang et al. (2011). The output of these models compares fairly well with observed salinity values, but their simplified
nature does not allow to identify the mechanisms of salt intrusion. Also, three-dimensional (3D) models have been used (e.g.
Xue et al. (2009); Bricheno et al. (2021); Bellafiore et al. (2021)). These models typically include a realistic geometry and a
wide range of physical processes that affect the salt transport. Disadvantages of these models include their high computational
costs, which makes extensive sensitivity studies computationally unfeasible. To fill the gap between simple 1D models and
3D numerical models, we develop here an idealised, partly analytical, width-averaged (2DV) model. This model is suitable
for a process-based analysis of salt intrusion mechanisms, as the model solves separately for different components of currents
and salinity, which makes it straightforward to compute different components of the salt transport. Furthermore, it has lower
computational costs than a 3D numerical model and is more flexible in terms of e.g. estuarine geometry.

To study the characteristics of salt intrusion in estuarine networks, the model will be applied to the Rhine-Meuse Delta (the
Netherlands) (RMD), which is formed by the outflow of the Rhine and Meuse rivers that discharge into the North Sea (Fig. 1).
The current geometry of the delta is to a large extent anthropogenic determined (Cox et al., 2021), with the large harbour of
Rotterdam situated in the seaward part. Earlier research on salt intrusion in this system by de Nijs and Pietrzak (2012), using
measurements and 3D model simulations, revealed that intertidal transport of salt between the channels occurs, but effects of
these transports on the tidally-averaged salinity were not quantified. Recently, salt intrusion in the RMD has received increased
attention. Kranenburg et al. (2022) showed that upstream salt transport is dominated by exchange flow in the seaward part and



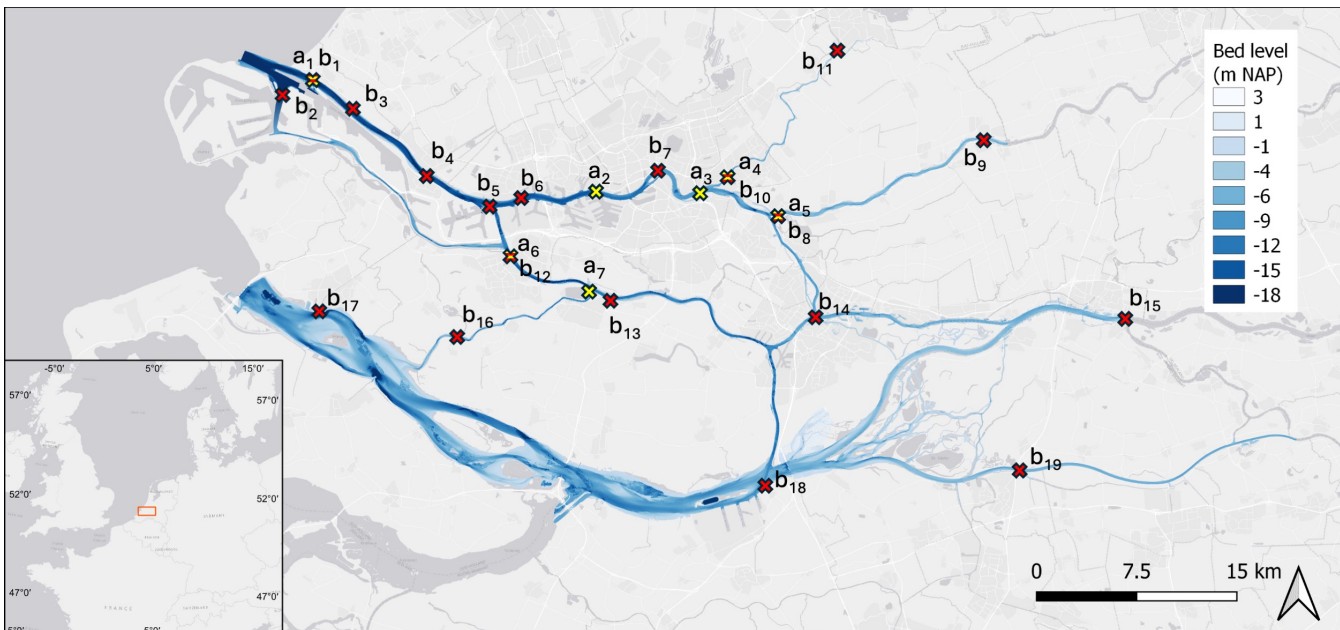

**Figure 1.** Map of the Rhine-Meuse Delta. The inset shows the location in Europe. The bed level is indicated in NAP, which is a measure of the mean sea level. The yellow crosses indicate salinity observation points ($a_1$-$a_8$) and the red crosses water level observation points ($b_1$-$b_{19}$). Locations where both these quantities are measured, or when these observation points are very close together, are indicated with both colours.

by tidal currents in the landward part of one of the major branches, but that during storm surges the relative importance of the different mechanisms changes. Wullems et al. (2023) point at the dependency of salinity on water level at one location in the network, using a data-driven method. None of these studies provide a network-wide analysis of salt intrusion.

To systematically investigate the dynamics of salt intrusion in estuarine networks, we formulated four specific aims: 1) Show that a calibrated idealised 2DV model successfully hindcasts hydrodynamics and salinity in an estuarine network. 2) Identify and quantify the different salt transport processes in an estuarine network, and highlight the differences with single-channel estuaries. 3) Quantify the time-dependent response to changes in discharge in an estuarine network. 4) Quantify the effects of changes to the depth of individual channels on salt intrusion in the major branches of an estuarine network.

The remainder of this manuscript is organised as follows. In Section 2, the model, its implementation in the RMD and the set-up of the simulations are described. Section 3 presents results focused on the four specific aims. Section 4 discusses the results and Section 5 describes the conclusions.





## 2 Model formulation

### 2.1 Domain

The domain mimics an estuarine network that has a number of width-averaged channels, which are connected to each other through a set of junctions. As a specific example, Fig. 2 shows the representation of the RMD network. There are four types of horizontal boundaries for the channels: river boundaries ($r_i$), weir boundaries ($w_i$), sea boundaries ($s_i$), and junctions ($j_i$). Salt enters through the sea boundaries. Fresh water enters (or leaves, e.g. in the case of the Haringvliet sluices) through the river and weir boundaries. The junctions provide the connections between the channels.

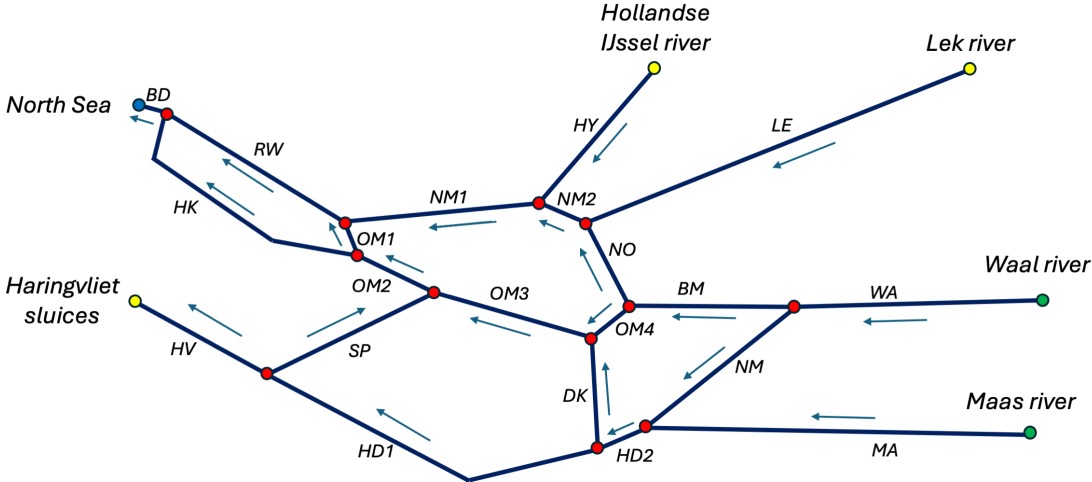

**Figure 2.** Representation of the RMD channel network in the model. Black lines indicate channels, blue arrows the direction of the local $x$-axis, red dots indicate junctions, yellow dots indicate weir boundaries, green dots indicate river boundaries, and blue dots indicate sea boundaries. Table A1 provides the meaning of the acronyms for the channels.

Each channel $i$ is divided in a number of segments $j$, which have different constant depths $H_{ij}$. Locations in the segments are characterised by a horizontal coordinate $x_{ij}$, which runs from $-L_{ij}$ to 0, and by a vertical coordinate $z_{ij}$, that runs from the bottom $z_{ij} = -H_{ij}$ to the water surface $z_{ij} = \eta_{ij}$, while the undisturbed water level is at $z_{ij} = 0$. By convention, $x_{ij}$ increases in the direction of the time- and cross-sectionally averaged flow for yearly averaged discharge conditions (indicated with arrows in Fig. 2). For clarity, the subscript $ij$ will be left out from now, and the equations shown are valid for each segment. The width of each segment is described as

$$b(x) = b(x = -L)e^{(x+L)/L_b}, \tag{1}$$





where $L_b$ is an e-folding length scale which describes the width convergence. The width is continuous at the boundaries of the
segments, but not necessarily at the junctions.

## 2.2 Equations of motion

The governing equations for hydrodynamics and salinity are described in Biemond et al. (2024a). In short, the model is width-
averaged, and on this domain momentum and continuity balances are solved for hydrodynamics and a mass balance is solved
for salinity. Water level $\eta$, horizontal and vertical velocity $u$ and $w$, and salinity $s$ are decomposed in tidally varying (indicated
with subscript $ti$) and subtidal (indicated with subscript $st$) components, and subtidal horizontal velocity and salinity are further
decomposed into a depth-averaged component (indicated with an overbar) and depth-dependent component (indicated with a
prime). This yields

$$\eta = \eta_{st} + \eta_{ti}, \quad u = \bar{u}_{st} + u'_{st} + u_{ti}, \quad w = w_{st} + w_{ti}, \quad s = \bar{s}_{st} + s'_{st} + s_{ti}. \tag{2}$$

The analytical solutions for the components of $\eta$, $u$ and $w$, as well as equations for the components of $s$ are given in Appendix B.
The shear of the subtidal current $u'_{st}$ depends on salinity, but not so for the depth-averaged subtidal current $\bar{u}_{st}$ and the tidal
current $u_{ti}$. The equation for $s_{ti}$ is solved analytically, but numerical methods are used to solve for $\bar{s}_{st}$ and $s'_{st}$, i.e. a Galerkin
method in the vertical and central differences in the horizontal are used. For time integration, a backward Euler scheme is
employed.

The horizontal salt transport $T$ in the model (which follows from the decomposition of currents and salinity as in Eq. 2),
integrated over the tidal cycle and cross-section, and after neglecting terms that contain $\eta$ (see Biemond et al. (2024a)), reads

$$T = \underbrace{Q\bar{s}_{st}}_{T_Q} + \underbrace{bH\overline{u'_{st}s'_{st}}}_{T_E} + \underbrace{bH\overline{(u_{ti}s_{ti})}_{st}}_{T_T} - \underbrace{bHK_{h,st}\frac{\partial \bar{s}_{st}}{\partial x}}_{T_D}. \tag{3}$$

Here, $K_{h,st}$ is the horizontal diffusion coefficient acting on the subtidal salinity and $Q$ is (tidally averaged) discharge. The
component $T_Q$ is the salt transport due to advection by the depth-averaged subtidal current, $T_E$ is the salt transport by the
exchange flow, $T_T$ is the salt transport due to time correlation of tidal flow and salinity and $T_D$ is the salt transport due to
horizontal diffusion. The net transport through a channel in equilibrium is called the salt overspill $T_o$. Note that $T_T$ could be
further decomposed into a contribution that involves only the depth-averaged tidal current and a contribution that is due to
the departure of the tidal current from its depth-averaged value. The values of vertical viscosity and diffusivity and horizontal
diffusivity are assumed to be constant throughout the entire domain. For bottom friction, a partial slip condition is applied at
the bed, with friction coefficient $S_f = \frac{2A_v}{H}$, where $A_v$ is the vertical viscosity and $H$ is the water depth. Note that viscosity,
diffusivity and friction coefficients are different when acting on the tidal or subtidal current and salinity (Godin, 1991, 1999).

Regarding the boundary conditions at river boundaries: subtidal discharge is prescribed, tides are assumed to vanish, and
subtidal salinity is set to the river salinity. Hence

$$bH\bar{u}_{st} = Q_{riv}, \quad \eta_{ti} = 0, \quad \bar{s}_{st} = s_{riv}, \quad s'_{st} = 0. \tag{4}$$





Here, $Q_{riv}$ and $s_{riv}$ are the river discharge and salinity, respectively. The condition that $\eta_{ti}$ vanishes is imposed numerically by extending the computational domain of the river channel, such that the tides are damped by internal friction before reaching the domain boundary.

At weir boundaries we prescribe subtidal discharge and use a reflecting boundary condition for the tidal flow, so the latter vanishes at such boundaries. For salinity, we impose conditions on the salt transport. When the discharge is away from the weir (which is usually the case), the depth-averaged transport equals a prescribed salinity $s_{weir}$ multiplied with the discharge. When the discharge is towards the weir (which is e.g. the case for the Haringvliet sluices in the RMD), the depth-averaged transport is set to the calculated salinity at the weir multiplied with the discharge. The vertical variations of the diffusive salt flux are set to zero, to avoid formation of a diffusive boundary layer (Biemond et al., 2024b). This reads

$$bH\bar{u}_{st} = Q_{weir}, \quad u_{ti} = 0, \quad T = Q_{weir}s_{weir}, \quad K_{h,st}\frac{\partial s'_{st}}{\partial x} = 0, \tag{5}$$

in which $Q_{weir}$ is the discharge at the weir, and $s_{weir}$ is prescribed in the case of the discharge being from the weir and the local (depth-averaged) salinity in case of a discharge towards the weir.

Conditions at sea boundaries are obtained as follows. The computational domain is extended, such that the adjacent sea is represented as the most downstream segment of the channel. This segment is characterised by a strongly increasing width. At the downstream boundary of this segment, salinity is prescribed to have the sea salinity. The water level at this boundary is chosen in such a way that at the estuary mouth the imposed tidal water level is reproduced. Subtidal water level is set to zero at the ocean boundary.

At the junctions, continuity of discharge and water level is imposed. These conditions apply to the subtidal quantities as well as to the tidally varying quantities. Regarding salinity, continuity of salt transport and salinity is used at every vertical level. To apply conditions to the tidal salinity, a boundary layer correction is employed. At the boundaries between the channel segments, the same conditions are applied. The specific expressions for these boundary conditions are described in Appendix C and the boundary layer correction for tidal salinity is described in Appendix D.

## 2.3 Model implementation for the RMD

The implementation of the domain of the RMD in the model, as sketched in Fig. 2, consists of 21 channels, 12 junctions, 1 sea boundary, 3 weir boundaries, and 2 river boundaries. The channels consist of one or two segments. The Haringvliet sluices connect the Haringvliet channel with the open sea, and are only opened to release water from the Haringvliet, such that no salt intrudes through this boundary. The widths and depths of each channel are presented in Table A1. Note that a coarse representation of the geometry is used so geometrical details, e.g. the harbour basins are not resolved. In the following, we will refer to the different channels by their acronyms as specified in Fig. 2.

The horizontal grid size is a few hundred meters, but differs per channel and is finer in regions where larger salinity gradients are expected. In the vertical, five Fourier modes are used for the Galerkin projection and the model time step (which only applies to subtidal quantities) is one day. The model is forced with prescribed time series of discharge at the two river and the three weir boundaries, and with the water level amplitude and phase of the dominant tidal constituent, which is the semi-diurnal



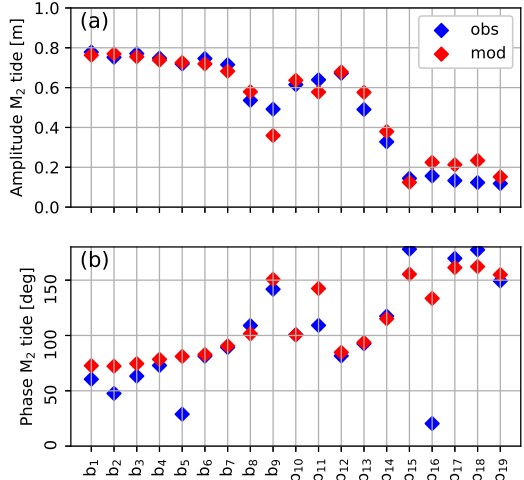

**Figure 3.** (a) Observed (in blue) and modelled (in red) amplitude of the $M_2$ water level for different points in the RMD, whose location is indicated in Fig. 1. (b) As (a), but for the tidal phase of the water level.

lunar $M_2$ tide (period 12 h, 25 m) for the RMD, at the North Sea boundary. For salinity, time-independent values are used, which are 0.15 g kg$^{-1}$ at the river and weir boundaries and 33 g kg$^{-1}$ at the North Sea boundary.

## 3 Results

### 3.1 Model-observation comparison

To address the first aim (model-observation comparison), suitable values of viscosity and diffusivity coefficients are needed. This is achieved by means of calibration: the modelled tidal water levels and salinities are compared with observed tidal water levels and salinities. There are 19 water level stations in the RMD, indicated with $b_1$-$b_{19}$ in Fig. 1, and 7 salinity observation points, indicated with $a_1$-$a_7$ in this figure. At a number of these locations, salinity observations are available at two or three
depths. The discharge, water level and salinity data of the RMD are accessible at waterinfo.rws.nl. The year 2022 is used as calibration period, since this is a year with a low Rhine discharge, which is a situation in which salt intrusion is relevant. For the hydrodynamic module, the value of $A_{v,ti}$, the vertical viscosity component acting on the tidal flow, is calibrated to get an optimal correspondence between the modelled and the observed $M_2$ water level variations, using the skill score as used by Davies and Jones (1996), which computes a cost function $f$ as

$$f = \sum_{k=1}^{N} \sqrt{(\hat{\eta}_k^{obs} - \hat{\eta}_k^{mod})^2 + 2\hat{\eta}_k^{obs}\hat{\eta}_k^{mod}\left(1 - \cos(\hat{\theta}_k^{obs} - \hat{\theta}_k^{mod})\right)}. \tag{6}$$



Here, $N$ is the number of observations, $\hat{\eta}_k^{obs}$ and $\hat{\eta}_k^{mod}$ are the observed and modelled amplitude of the water level of the dominant tidal constituent, respectively, and $\hat{\theta}_k^{obs}$ and $\hat{\theta}_k^{mod}$ are the observed and modelled phases of the tidal water level, respectively.

Subtidal vertical eddy viscosity $A_{v,st}$, subtidal vertical and horizontal eddy diffusivity $K_{v,st}$ and $K_{h,st}$ and tidal vertical diffusivity $K_{v,ti}$ are calibrated by minimizing the root-mean-squared error (RMSE) between observed and modelled time series of subtidal salinity at the observations points. A gradient descent algorithm is employed to find the optimal values of these parameters. After the calibration, the model performance regarding subtidal salinity is quantified by calculating the Nash-Sutcliffe efficiency ($NSE$) (Nash and Sutcliffe, 1970), which reads

$$NSE = 1 - \frac{\sum_{k=1}^{N}(s_k^{obs} - s_k^{mod})^2}{\sum_{k=1}^{N}(s_k^{obs} - <s_k^{obs}>)^2}, \tag{7}$$

where $s_k^{obs}$ and $s_k^{mod}$ are observed and modelled salinity, respectively, $<\cdot>$ indicates a time-averaged quantity, $NSE = 1$ indicates perfect agreement and $NSE = 0$ means an that the model error is equal to the variance of the observed time series.

The amplitudes and phases of the $M_2$ component of the water level in observations and the calibrated model are shown in Fig. 3. A high level of agreement is obtained with $A_{v,ti} = 0.025$ m$^2$s$^{-1}$. The amplitude in the main channels is well resolved ($b_1 - b_7$, $b_{12} - b_{14}$). In observation point $b_9$ (the LE channel), the amplitude is underestimated by 27%, and in the southern part ($b_{16} - b_{19}$), the amplitude is overestimated by 55% averaged over the four points. There are two outliers in terms of tidal phases (at points $b_5$ and $b_{16}$). The observed phase at point $b_5$ differs in such a way from the phase at the neighbouring points that probably an error in the raw data is responsible for the mismatch here. In point $b_{16}$ the tides are affected by the operation of the Haringvliet sluices, which is not resolved in the model.

The best agreement with modelled salinity is found using $A_{v,st} = 0.0024$ m$^2$s$^{-1}$, one order of magnitude lower than the vertical viscosity for tidal flow. Regarding vertical diffusivities for tidal and subtidal salinity, the best agreement is found when using $K_{v,st} = \frac{A_{v,st}}{Sc}$ and $K_{v,ti} = \frac{A_{v,ti}}{Sc}$, with $Sc = 2.2$, the Schmidt number. The horizontal subtidal diffusivity $K_{h,st} = 275$ m$^2$s$^{-1}$, which is rather high compared to what was found for single-channel estuaries (Biemond et al., 2024a). The reasons and implications of this will be discussed in Section 4.

Modelled and observed salinity in the RMD at four selected observation points is shown in Fig. 4. We refer to Fig. S1 for the comparison at the other observations points. The overall Nash-Sutcliffe efficiency (Eq. 7) is 0.67, which classifies the model performance as satisfactory (Moriasi et al., 2015). Differences between the model and observations primarily concern an underestimation of the variability of the salt field. This is due to the fact that our model does not account for the spring-neap tidal cycle, overtides and subtidal and intratidal water level fluctuations at the sea boundary driven by remote winds, which have a strong influence on the variability of the salinity. The model-data agreement is consistent at different depths for most stations where observations are available at multiple depths (Fig. S1), but the salinity in station $a_1$ is underestimated by 3.0 g kg$^{-1}$ at $z = -9.0$ m averaged over first two months of the year, i.e. during high discharge conditions, while at $z = -2.5$ m no bias is present (Compare Fig. S1d with Fig. S1f). This indicates that the relation between discharge and vertical salinity gradients in the model is not well resolved, which probably is due to the use of a constant vertical viscosity and diffusivity. In Fig. 4b, we see an underestimation of 0.6 g kg$^{-1}$ averaged over the summer months, while in Fig. 4d, which is only a few kilometres from



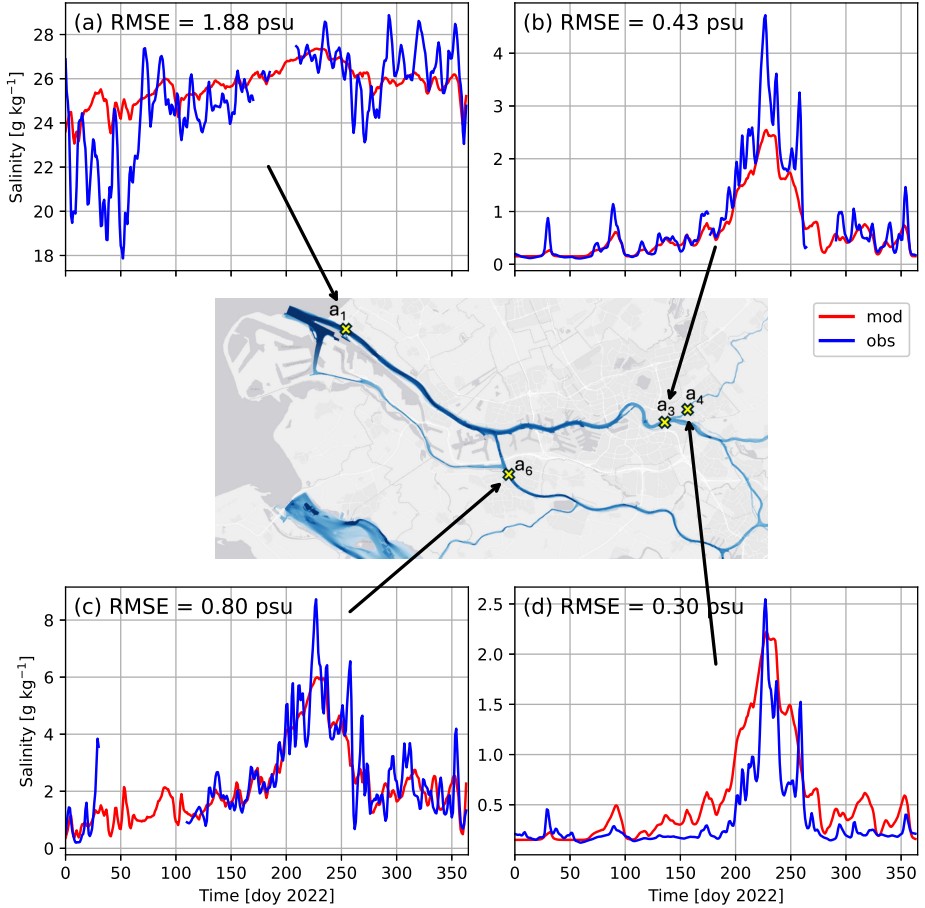

**Figure 4.** Observed (in blue) and modelled (in red) tidally averaged salinity time series for 2022 at four locations in the RMD: (a) $a_1$ ($z = -9.0$ m), (b) $a_3$ ($z = -2.5$ m), (c) $a_6$ ($z = -2.5$ m), and (d) $a_4$ ($z = -4.0$ m).

this location, the salinity is overestimated by 0.4 g kg$^{-1}$ during this period. This probably originates from the large value of $K_{h,st}$, which smoothens horizontal salinity gradients, but also the exclusion of other tidal constituents besides M$_2$ play a role, as tidal flow is very important to salinity at point $a_4$, which will be shown in the next section.

## 3.2 Salt intrusion mechanisms

To quantify the mechanisms of salt transport in an estuarine network (the second research aim), an equilibrium simulation is performed with the calibrated model. To this end, the model is forced with discharges that represent low discharge conditions, as this is the situation in which salt intrusion is the most relevant. For the Haringvliet sluices, Hollandse IJssel and Lek rivers we set the discharge to zero, for the Waal river we use $Q = 650$ m$^3$s$^{-1}$ and for the Maas river $Q = 50$ m$^3$s$^{-1}$. These values equal approximately the minimum observed discharges in the summer of 2022.



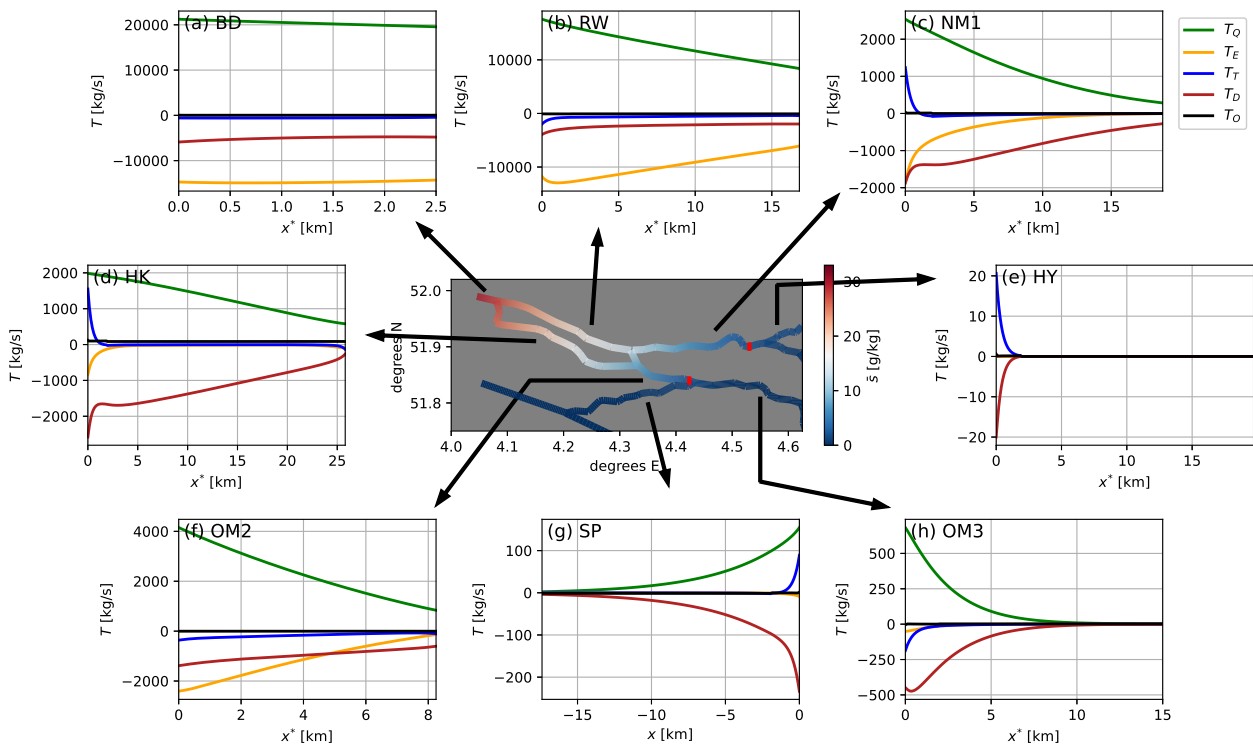

**Figure 5.** Center panel: depth- and tidally averaged salinity in the RMD for low discharge conditions. The red bars in this figure indicate the location of the 2 g kg$^{-1}$ isohaline. (a)-(h) The different components of the salt transport in equilibrium, following the decomposition of Eq. 3, versus $x$ or $x^*$, where we have defined $x^* \equiv -x$ for plotting purposes.

The distribution of net water transport and subtidal salinity in different channels in the RMD are presented in Fig. S2; the

205 associated components of the salt transport in different channels in the RMD for low discharge conditions are shown in Fig. 5. It turns out that close to the sea boundary (BD, RW), salt transport related to the exchange flow $T_E$ is the most important contribution to the salt transport, further inland its relative contribution decreases. This spatial pattern is explained by the fact that close to the sea boundary, depth is large, but further inland, the RMD becomes shallower (e.g. $H = 16$ m for BD and RW, but $H = 11$ m for NM1 and $H = 10.2$ m for OM3). The strength of the exchange flow $u'_{st}$ scales with $H^3$ (Eq. B1c), and

210 variable $s'_{st}$, which is mainly generated by this flow, thus decreases when depth decreases (Fig. S3). Salt transport component $T_E$ is the product of those two and therefore strongly depends on depth.

The salt transport component due to horizontal diffusion $T_D$ becomes dominant in most areas where $T_E$ declines (e.g. NM1, SP, OM3). This is because the dependence of $T_D$ on depth is weaker (only through a decrease of the cross-section). The component $T_T$ is small in most of the network. To explain this, we compute the phase difference between depth-averaged

tidally varying flow and salinity, which is very important to the magnitude of $T_T$ (see Eq. 3). A phase difference of more



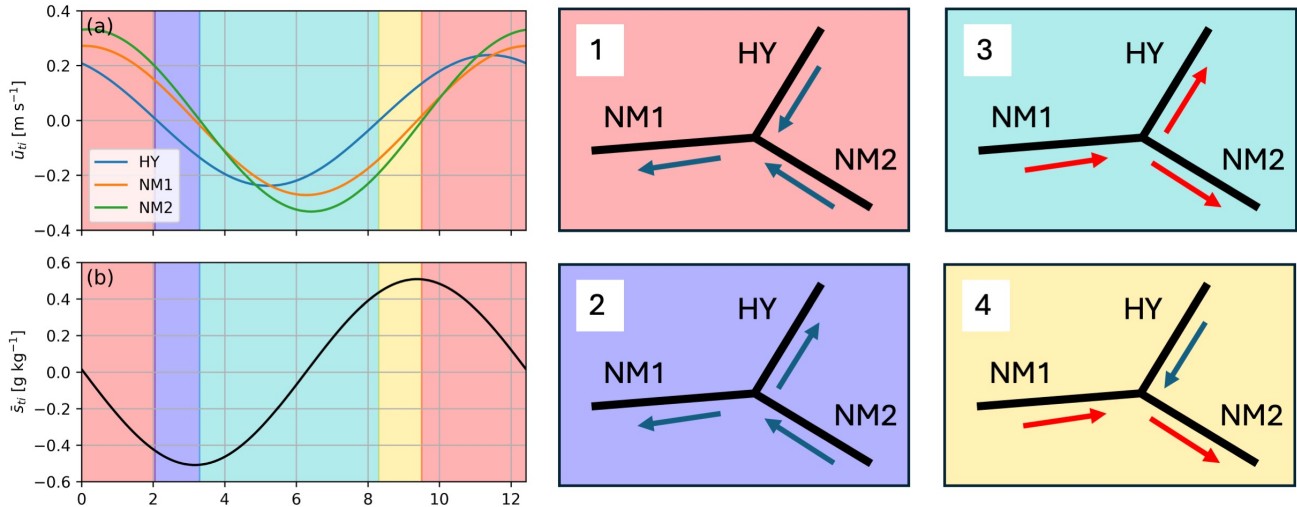

**Figure 6.** (a) Depth-averaged tidal velocity $\bar{u}_{ti}$ within a tidal cycle at the junction of the Hollandse IJssel (in blue), Nieuwe Maas 1 (in orange) and Nieuwe Maas 2 (in green). (b) As (a), but for depth-averaged tidal salinity $\bar{s}_{ti}$. Panels 1-4: Visualisation of situation during different phases of the tide. The background colors correspond to the colors of panel a-b. The arrows indicate the direction of the current and its color indicates whether the water is relatively fresh (blue) or relatively saline (red).

(less) than 90 degrees is associated with an upstream (downstream) salt transport. In our model context, this phase difference is generated by two mechanisms: vertical advection of subtidal salinity within the tidal cycle (Biemond et al., 2024a), and interaction with other channels around junctions. The first mechanism requires the presence of tidally-averaged vertical salinity gradients, which mainly are created by the exchange flow. This gives, for example, a phase difference of about 95 degrees in

the RW (Fig. S4), which yields $T_T \approx 500\,\mathrm{kg\,s^{-1}}$, about 15 times smaller than $T_E$. Note that a $T_T$ of this strength would be the dominant salt transport in channels further upstream, e.g. OM3. In the channels where exchange flow is weaker, the induced phase difference is also smaller, and therefore $T_T$ is nowhere a dominant component of the salt transport, except around some junctions. Vertical variations of the tidally varying flow and salinity are not considered for this explanation, because their contribution to salt transport is negligible, as explained in Biemond et al. (2024a).

Around junctions as the SP-OM2-OM3 or the BD-RW-HK, the distortion of the phase difference from 90 degrees increases strongly, e.g. around the seaward boundary of the HK, NM1 and HY, where the minimum phase difference is about 60 degrees (Fig. S4). Here, $T_T$ also reverses sign, i.e. it becomes seaward, along the salinity gradient. Note that for each junction in only one connected channel the phase difference is below 90 degrees. To illustrate this behaviour, we study the case of the junction of the HY, NM1 and NM2 in detail, but similar reasoning applies to the other junctions. Fig. 6 visualises tidal currents and

salinity around this junction. Panel a shows that $\bar{u}_{ti}$ in the HY leads the currents in the other two channels by about 1.5 hours. The consequences for the salt transport are made clear by studying the different phases of the tide. When the currents in all



channels are positive, i.e. the red shaded area and panel 1, relatively fresh water from upstream is transported downstream, and salinity at the junction decreases (panel b). At some point, the current in the HY will change direction before the currents in the other channels change direction (the blue shaded area and panel 2). In this phase, the HY will receive freshwater from upstream, i.e. from the NM2. After about 1.5 hours, currents in the other channels will also change direction (the light blue area and panel 3), and saline water from downstream is imported into the channels. However, when salinity at the junction reaches its maximum value, $\bar{u}_{ti}$ in the HY has changed direction (the yellow area and panel 4), and this salinity does not reach the HY. The tidally averaged transport $T_T$ is upstream in the main channel (NM1 and NM2), while it is downstream in the side channel (HY).

Moreover, the HY (and also the LE) is a special case in a sense that it does not receive discharge from upstream and therefore $T_Q$ is everywhere zero in this channel. The absence of discharge, which is the main source of buoyancy, also implies that (tidally averaged) vertical salinity gradients are almost absent: the top-bottom salinity difference is 0.09 g kg$^{-1}$ at 1 km from the junction in the HY. This implies that $T_E$ is weak, i.e. $\frac{T_E}{T_D} = 5 \cdot 10^{-4}$ at this location. A similar mechanism plays a role in the HK, which is shallower than its neighbouring channels, therefore receives less discharge (71 m$^3$s$^{-1}$, about 9 times smaller than the RW) and consequently the channel-averaged contribution of $T_E$ to the upstream salt transport in the HK is only 4% (Fig. 5d).

Finally, the absolute value of the total salt transport $|T| = 80$ kg s$^{-1}$ in the HK, OM1 and RW, indicating the presence of salt overspill in these channels. Overspill occurs when salt originating from one channel is transported downstream with the subtidal current $\bar{u}_{st}$ through another channel. In Fig. S2 the direction of this transport is indicated. For the range of values of the parameters investigated in this study, this only occurs in the RW-OM1-HK channel system in the RMD. Under the low discharge conditions studied here, its contribution to the total salt transport in the HK is 13%.

### 3.3 Changes in discharge

To examine the response of salinity in the network to changes in discharge (the third research aim), the model is forced with discharge time series that represent the transition from reference discharge conditions to low discharge conditions and vice versa. For the Haringvliet sluices and the Hollandse IJssel river, discharge is set to zero. In the reference situation, the Lek, Waal and Maas rivers are forced with their long-yearly average discharge, which are 490, 1500 and 230 m$^3$s$^{-1}$, respectively. To simulate the response to a transition to low discharge conditions, the reference discharges are multiplied with a factor, ranging from 0.95 to 0.2 for the different scenarios. In this manner, the fraction of the total discharge received per channel does not change. The simulation is continued until equilibrium has been reached. To simulate the response to an increase in river discharge, discharge is increased from the lowered value to the yearly average values, and again the simulation is continued until equilibrium has been obtained. Note that these simulations do not reproduce the current water distribution in the RMD during a drought, because discharge in the three rivers does not change with the same ratio, due to water management and local differences in drought intensity. Moreover, the Haringvliet sluices are opened when the discharge is high, closed for low discharge, and partially opened in between.





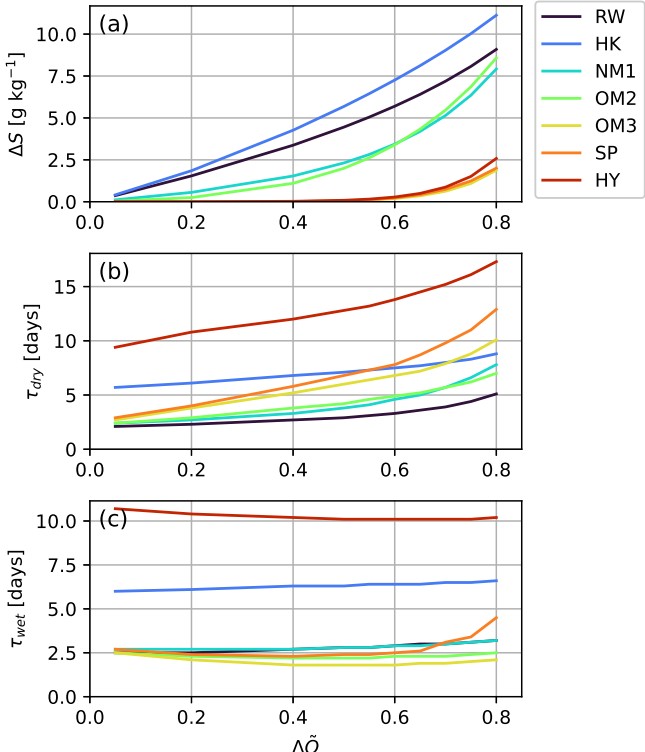

**Figure 7.** (a) Change in channel-averaged salinity $\Delta S$ of different channels as a function of change in discharge $\Delta \tilde{Q} = \frac{Q_{ref} - Q_{dry}}{Q_{ref}}$, where $Q_{ref}$ is the yearly average discharge and $Q_{dry}$ the decreased discharge. (b) As (a), but for the response time $\tau_{dry}$ due to a decrease in river discharge, i.e. a transition from $Q_{ref}$ to $Q_{dry}$. (c) As (b), but for the response time $\tau_{wet}$ due to an increase in river discharge, i.e. a transition from $Q_{dry}$ to $Q_{ref}$.

For quantification of the salinity response to a change in discharge, studies for single-channel estuaries have analysed the evolution of a certain isohaline, typically the 2 g kg$^{-1}$ isohaline (Chen, 2015; Monismith, 2017). However, in a network this is not generally possible, as isohalines are not bound to a certain channel. Therefore, an alternative measure for the salt intrusion in a channel is used for response times, which is based on the channel-averaged salinity $S$, defined as $S = \frac{\int_V s_{st} dV}{V}$, where $V$ is the volume of the channel. The response time is then defined when 90% of the change in $S$ has occurred. We indicate the

time it takes to respond to a low discharge with $\tau_{dry}$ and the time it takes to respond to a high (i.e. the reference) discharge with $\tau_{wet}$.

The changes in mean salinity and the response times of individual channels in the RMD after an increase and decrease in discharge of the Lek, Waal and Maas rivers are shown in Fig. 7. A larger change in discharge is associated with a larger change in mean salinity, as expected. The absolute changes in salinity are larger for channels close to the sea (RW, HK), but

smaller for channels close to the limit of the salt intrusion (SP, HY, OM3). Regarding response times, $\tau_{dry}$ increases for larger





changes in discharge ($\tau_{dry}$ associated with an 80% change in discharge is 61% larger than $\tau_{dry}$ of a 5% change for the channels considered in Fig. 7), but $\tau_{wet}$ is not sensitive to the magnitude of the change in discharge ($\tau_{wet}$ associated with an 80% change in discharge is only 9% larger than $\tau_{wet}$ of a 5% change for the channels considered in Fig. 7). This implies that $\tau_{dry}$ and $\tau_{wet}$ differ little for small changes in discharge, but that $\tau_{dry}$ exceeds $\tau_{wet}$ for larger changes in discharge.

Following Biemond et al. (2022), who also found this result for single-channel estuaries, the explanation is that $\tau_{dry}$ equals change in salt content, which scales with the change in salinity, divided by salt import due to $T_E$ and $T_D$, which depend on the salinity gradient during the adjustment. The change in salinity and salinity gradient are in another manner related to the change in discharge and therefore $\tau_{dry}$ depends on the change in discharge. On the other hand, $\tau_{wet}$ is mostly determined by the change in salt content divided by the export of salt due to $T_Q$, which both scale linearly with salinity and therefore their

ratio does not depend on the change in discharge.

The channels closer to the estuary mouth, i.e. those with a high salinity (RW, NM1), have a smaller $\tau_{dry}$ than the channels with a low salinity (HY, SP). This follows from the fact that channels with a higher salinity also have larger horizontal and vertical salinity gradients (Figs. S3 and S5), which increases the magnitude of the salt import by $T_E$ and $T_D$, and thus reduces $\tau_{dry}$.

The HK and HY have larger response times than their neighbouring channels. Focussing first on the HK, for which $\tau_{dry}$ is a factor two higher than in the RW, despite the salinity and salinity gradients being comparable. To explain this, we calculated the total change of salt content in this channel, which equals the channel volume $V$ multiplied with the change in its average salinity $\Delta S$, divided by its salt import in equilibrium at the downstream boundary $T_r = \frac{V\Delta S}{T_E+T_D+T_T}$, which is a lower bound of $\tau_{dry}$. The salt exchange at the upstream boundary is a factor 6 smaller in this channel (Fig. 5d) and therefore not taken

into account here. We find $T_r = 18.4$ h for the HK and e.g. $T_r = 4.0$ h for the NM1 and $T_r = 3.0$ h for the OM2. Thus, the ratio between salt exchange with the neighbouring channels to the total change in salt content in the HK is smaller than for neighbouring channels, which is because this channel receives little discharge, and this explains the larger $\tau_{dry}$. The same reasoning holds for the fact that $\tau_{wet}$ of this channel is larger than in its neighbours, as in equilibrium the salt import equals the salt export, and therefore the salt export is also relatively small.

In the HY, $\tau_{dry}$ is about a factor two higher than that of the other channels. In channels without net water transport like the HY, equilibrium with the forcing conditions will be reached when the horizontal salinity gradient is zero in the channel interior, i.e. when the salinity at the upstream boundary equals the salinity just outside the boundary layer around the junction. This means that the salt front has to intrude over the full length of the channel, which makes that the response time is larger than for channels with net water transport. Moreover, only salt transport component $T_D$ (see Section 3.2) imports salt in this channel

during the adjustment process, while $T_T$ exports salt around the junction (Fig. 5e). The high $\tau_{dry}$ of the HY, and more general of channels without discharge, is thus explained by the fact that salt import by $T_E$ is absent and that salt has to intrude over the full length of the channel.

Fig. 7c shows that $\tau_{wet}$ is also high, because there is no discharge through the channel which flushes the salinity (through $T_Q$). Instead, salt is removed by $T_D$ and $T_T$. Like for the other channels, $\tau_{wet}$ is smaller than $\tau_{dry}$ in the HY, but different

mechanisms play a role here than for most other channels. First, $T_T$ has the same sign as $T_D$ in the boundary layer around the





junction when salinity in the channel decreases, so salt export during a decrease in salinity is faster than salt import during an increase in salinity. Second, the salinity in the channels where it is connected to responds faster to an increase in discharge than to a decrease in discharge (compare $\tau_{wet}$ and $\tau_{dry}$ of NM1).

### 3.4 Changes in depth

Effects of changes to the depth of individual channels on salt intrusion in the major branches (the fourth research aim) are quantified from analysing output of a set of equilibrium simulations which have the same forcing conditions as the simulation for the second research aim. In each of these simulations, the depth of one of the channels in the network is increased or decreased by 25%. The impact on the salt intrusion is quantified by means of the distance $X_2$ from the RW-NM1-OM1 junction to the subtidal 2 g kg$^{-1}$ isohaline in the Nieuwe Maas (consisting of channels NM1 and NM2) and in the Oude Maas

(consisting of channels OM1, OM2, OM3, OM4), the two major branches of the network. The locations of these isohalines are shown in the centre panel of Fig. 5. These quantities can be converted into distance to the sea boundary by adding the length of the BD and that of RW, which is 19.3 km in total.

Salt intrusion lengths $X_2$ of the Oude Maas and Nieuwe Maas for changes to the depth of individual channels are shown in Fig. 8a. In contrast to the simulations for Section 3.3, the net water transport distribution changes when the depth changes, and

the values for the major branches are presented in Fig. 8b. The responses of the salt intrusion lengths range from no detectable change in $X_2$ to a maximum change of 12 km. In the following, we will explain the mechanisms that are responsible for the different responses.

Changes in depth of the most seaward channels (BD and RW) affect the extent of the salt intrusion (increases when deepening, decreases when shoaling), because upstream salt transport in these channels is dominated by $T_E$ (Fig. 5a), which depends

strongly on the local depth (Section 3.2). Changing the depth of the RW has a larger effect on $X_2$ than changing the depth of the BD because the RW is longer. The magnitude of the change in $X_2$ due to a decrease $+\Delta H$ in depth exceeds the change after an increase $-\Delta H$. Scaling relations between depth and extent of the salt intrusion in single-channel estuaries (Monismith et al., 2002) predict a scaling with $H^3$ when $T_E$ is the dominant salt import mechanism, which implies that, in contrast with what we find, an increase in depth would result in a larger absolute change in $X_2$ than a decrease.

To understand that this is different in a network, here for the RW, a simple model is constructed, which mimics the RW as a single-channel estuary, but with two segments, and assumes a balance between $T_Q$ and $T_E$ (the Chatwin (1976) balance), which are the dominant salt transport components in the RW (Fig. 5b). The details of the model are in Appendix E. In this case, analytical solutions for the salt intrusion length as a function of depth are obtained. Two scenarios are considered: in the first, the depth of the entire channel is changed. In the second, only the depth of the downstream 20 km long segment of the channel

is changed. The results are shown in Fig. 8c. When the depth of the entire channel is changed (the green line), an increase in depth has a larger impact than a decrease in depth, which follows from the earlier mentioned $H^3$ scaling (Eq. E2). However, when the depth of only the downstream part of the channel is changed (the purple line), a deepening $+\Delta H$ has a smaller effect than shoaling $-\Delta H$. This is because when the depth is increased, salt intrusion length increases, and only a limited part of the salt intrusion experiences a larger depth and the associated increase in salt transport. In case of shoaling, the salt intrusion



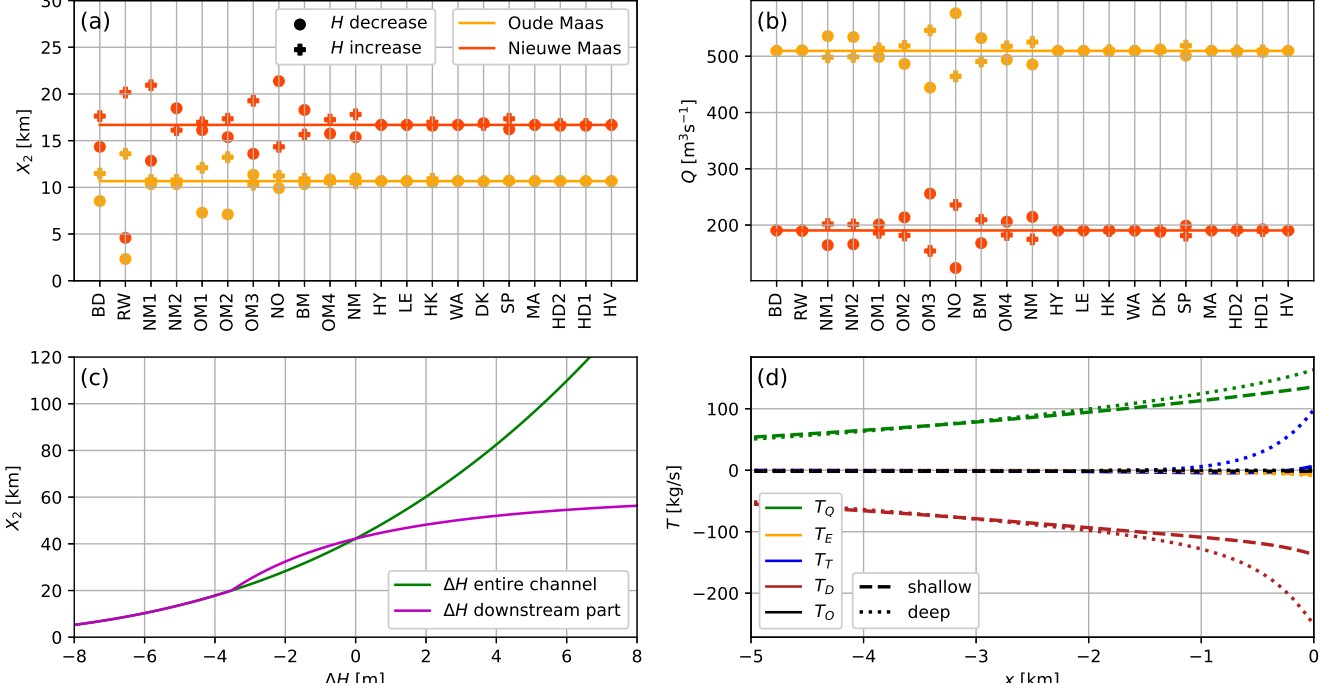

**Figure 8.** (a) The salt intrusion length $X_2$, defined as distance from the RW-NM1-OM1 junction to the 2 g kg$^{-1}$ isohaline in the Oude Maas (yellow), Nieuwe Maas (red), versus changes in depth of different channels. The dots indicate a decrease in depth, and the plus markers indicate an increase in depth. The lines indicate $X_2$ with the current geometry. (b) As (a), but for the net water transport through OM1 (yellow) and NM1 (red). (c) Salt intrusion length $X_2$ of a single-channel estuary that consists of two segments, as a function of change in depth $\Delta H$ of the entire estuary (green) or change in depth of the downstream 20 km (purple), around a reference depth of 16 m (the depth of the RW), using Eqs. E4 and E5. (d) As Fig. 5g, but for a shallowed HD1 (dashed line) or a deepened HD1 (dotted line) and zoomed in on the 5 km of the channel adjacent to the SP-OM2-OM3 junction.

length decreases and a relatively large part of the salt intrusion experiences a smaller depth and the related decrease in salt transport. As the RW and BD are within the range of salt intrusion, increasing their depth has a smaller effect than decreasing their depth.

For changes to the depth of channels which are part of major branches Oude Maas and Nieuwe Maas, two competing mechanisms play a role. Deepening (shoaling) will increase (decrease) the magnitude of $T_E$. Simultaneously, the net water

transport through the branch will increase (decrease), due to a decrease (increase) in friction, causing an increase (decrease) in $T_Q$. Because of volume conservation, the magnitude of the change in net water transport is equal for the other major branch, but with an opposite sign. The change in friction when changing the depth also affects the tidal currents, but the changes in salt transport component $T_T$ are negligible. The interplay of these mechanisms makes that when deepening (shoaling) channels that are part of the major branches and within the limit of the salt intrusion, i.e. the NM1, OM1, and OM2, salt intrusion



increases (decreases). This is because locally the magnitude of $T_E$ increases (decreases), which dominates over the increase (decrease) in $T_Q$. In the other major branch, salt intrusion increases (decreases) as well, because net water transport decreases (increases) (Fig. 8b). On the other hand, changing the depth of channels that are part of the major branches but outside the limit of the salt intrusion (NM2, OM3) does not affect $T_E$ directly, but only $T_Q$. This makes that an increase (decrease) in depth decreases (increases) salt intrusion in the major branch where the channel is part of, and simultaneously increases (decreases)

salt intrusion in the other major branch. Changes in net water transport also explain the response of channels that are important for the distribution of the Waal river discharge, i.e. the NO, BM, OM4 and NM. Especially the NO has a large effect on salt intrusion in the Nieuwe Maas, as this channel exerts a strong control on the amount of discharge from the Waal river that reaches the Nieuwe Maas. The salt intrusion in the Nieuwe Maas is more sensitive to absolute changes in net water transport than the Oude Maas, because the net water transport through the Nieuwe Maas is about 2.5 times smaller than that of the Oude

Maas (Fig. S2).

Changes in the depth of the other channels in the RMD (in Fig. 8a HY and all to its right) have little impact on $X_2$ in the Nieuwe Maas and Oude Maas (changes are less than 1 km). There are however two cases, which have an interesting local effect. In the reference simulation, salt overspill occurs in the HK ($T_o = 80$ kg s$^{-1}$) and salt from the Oude Maas is transported seawards through the HK (Figs. 5d and S2). When decreasing its depth, the magnitude of the salt transport due to salt overspill

increases ($T_o = 540$ kg s$^{-1}$), but when increasing the depth, the sign of $T_o$ reverses and the HK becomes a source of salt for the Oude Maas ($T_o = -250$ kg s$^{-1}$). The consequence of this change in salt transport for the salt intrusion length is only a few hundred meters, because this salt transport is relatively weak compared to the amount of salt transported from the RW into the Oude Maas. Secondly, increasing or decreasing the depth of HD1 has little impact on $X_2$ in the Oude Maas and Nieuwe Maas, but it affects the extent of the salt intrusion into the SP. The distance of the 0.5 g kg$^{-1}$ isohaline in the SP to the junction with

the OM2 and OM3 is 6.3, 5.2 and 4.8 km for the shallow, reference and deep HD1, respectively, so shoaling the HD1 affects salt intrusion more than deepening. To understand this, we calculated the change in salt transport components $T_Q$ and $T_T$ (the flow associated with these components of the salt transport directly changes in the SP due to a change of the depth of the HD1). The difference in $T_Q$ is about 30 kg s$^{-1}$ between the shallow and deep HD1. Fig. 5g shows that for the reference depth $T_T$ in the SP close to its seaward boundary is positive, in other words, salt transport due to tidal currents is directed towards the

SP-OM2-OM3 junction. Fig. 8d shows that this situation persists for a deep HD1, but when decreasing the depth of HD1, the magnitude of $T_T$ decreases, due to a shift in phases of the tidal currents of the different channels around this junction. The difference in $T_T$ at the junction is about 90 kg s$^{-1}$, three times larger than the change in $T_Q$, indicating that the change in tidal currents dominates the change in salt intrusion into the SP when deepening the HD1.

## 4  Discussion

When our idealised network model is applied to the RMD and calibrated against observations, it has a Nash-Sutcliffe efficiency of 0.67. This indicates that the model satisfactory represent observed salinity values, despite the fact that the descriptions of





the physics, forcing conditions and bathymetry are rather simple. The gradient descent algorithm to find optimal values of the calibration coefficients is crucial to obtain a satisfactory skill.

The effects of the non-resolved aspects of the salinity dynamics on the salt transport are via the calibration mostly projected
on horizontal salt dispersion. A value of $K_{h,st} = 275$ m$^2$s$^{-1}$ is found, and with this value $T_D$, the tidal dispersion, is the dominant salt transport component in parts of the RMD, in particular around the limit of the salt intrusion. However, since most of the observations points are in the larger channels of the network, this value of tidal dispersion mainly represents the unresolved salt transport in these channels. A hint that in the smaller channels of the network this constant value is too large follows from the described difference in model-data agreement between point $a_3$ and $a_4$ (see Section 3.1). To resolve
the spatial variation in salt transport by tides, a spatially varying $K_{h,st}$ could be employed, but finding relations between e.g. channel geometry and this quantity is not straightforward (Banas et al., 2004; MacCready, 2007; Aristizábal and Chant, 2015). A better method would be to resolve a larger fraction of the salt transport by tidal currents explicitly, i.e. incorporate more of the tidal salt transport in $T_T$ instead of $T_D$. For this, aspects of the model could be modified in a future study. The most promising directions are inclusion of additional mechanisms which generate a phase difference between tidal currents and salinity, e.g.
overtides, lateral effects (Scully and Geyer, 2012) or tidal trapping in side embayments (Okubo, 1973; Geyer and Signell, 1992; MacVean and Stacey, 2011). Also an improvement in the description of the salt transport is expected when missing forcing conditions are added, e.g. water level fluctuations at subtidal and intratidal time scales at the sea boundary (Kranenburg et al., 2022) and water extractions, which are of order $100$ m$^3$s$^{-1}$ (Huismans et al., 2024), about one seventh of the total discharge in dry conditions.

We found that response times vary between the channels and attributed this to differences in the salinity gradients, volume/-transport ratios of the channels, and the presence of net water transport in the channels. Observations show that response times to changes in forcing are observed to be 10-14 days for the HY (Laan et al., 2021), in line with what we find. In another estuarine network, the Po Delta, differences in response times between the channels are minor (Bellafiore et al. (2021) Fig. 5a). This is because in that delta all channels receive a considerable fraction of the river discharge and differences in channel depth are
also relatively small compared to the RMD. Therefore, dynamics of e.g. side branches, like the HY in the RMD, do not play a role in the Po Delta. Certain aspects of the theory of response of single-channel estuaries to changes in river discharge can be transferred to estuarine networks, e.g. the reason for the asymmetry in the response to an increase or decrease in discharge. However, a direct comparison with single-channel estuaries is hindered by e.g. differences in definitions of response times for reasons that were explained in Section 3.3.

We showed that in a network, the response of the salt intrusion to changes in depth is sensitive to which channel is changed. Salt intrusion can decrease locally when parts of an estuarine network are deepened, due to a redistribution of the river water. This is different for single-channel estuaries, for which it is found that salt intrusion increases when the depth increases (Ralston and Geyer, 2019; Siles-Ajamil et al., 2019; Kolb et al., 2022). Liu et al. (2019) also reported effects on salt intrusion when upstream in a network (the Pearl River Delta) discharge changes due to alterations of the depth. For the Yangtze Estuary, Zhu
et al. (2006) reported that salt intrusion increased in multiple channels when the depth of one the channels was increased, in line with what we find for the RMD. Huismans and Plieger (2019) showed, using a 3D model of the RMD, that a shallower



Oude Maas causes a decrease of salt intrusion in the Nieuwe Maas as well as in the Oude Maas. However, when winds and storm surges are accounted for, a more complicated response to changes in depth occurs. They hypothesised that changes in phases of the currents around the junction could be important to the response to changes in depth; we showed that they indeed can play an important role. For a deepening of the RW, Laan et al. (2023) found that salt intrusion in the entire RMD changes, but when deepening the LE, the effect remains local, which is again in line with what we find.

## 5 Conclusions

The main findings, based on simulations with our idealised, partly analytical, 2DV model of an estuarine network, are the following:

1. The network model, when calibrated, has a satisfactory agreement with observations, despite missing potentially important aspects of the salinity dynamics. The reason for this is that the salt transport associated with these aspects projects well on the calibrated processes, especially on the horizontal salt dispersion. A consequence is that local variations in the salt transport due to tidal currents are only crudely resolved.

2. Salt transport components related to different mechanisms vary in relative importance within a network, due to differences in water depth, discharge, phases of the tidal currents and characteristics of the salt field itself. In the deeper channels of the RMD, upstream salt transport by exchange flow is dominant, but in the shallower parts tidal dispersion takes over. Salt transport by tidal currents can be directed seawards around a junction when the currents of the different channels are out of phase.

3. Channels closer to the estuary mouth respond faster to changes in discharge than those further upstream. The response to an increase in discharge is faster than the response to a decrease, in agreement with theory of single-channel estuaries. Channels which receive little discharge have large response times, especially when the net water transport is zero.

4. The impact of changes to the depth of an individual channel in a network on the extent of salt intrusion in the major branches depends on where the change takes place. The impact is determined by the direct, local effect on e.g. the strength of the exchange flow, and the interaction with other channels, which is related to changes in the distribution of discharge and tidal currents. Scaling relationships developed for single-channel estuaries are found to be invalid for depth changes in an estuarine network. Salt intrusion in a channel with lower discharge is more sensitive to changes in depth than a channel with higher discharge. The phase differences of tidal currents around a junction are sensitive to changes to the depth and this sensitivity can be the cause of changes in salt intrusion when the depth of a channel in another part of the network is changed.



*Code and data availability.* Software used in this study is made available online (Biemond, 2024). More recent versions of the model, called IMSIDE (Idealised Model of Salt Intrusion in Deltas and Estuaries), are available at https://github.com/nietBouke/IMSIDE. Used data can be downloaded from waterinfo.rws.nl.





## Appendix A: Geometry of the Rhine-Meuse Delta

**Table A1.** Geometry of the Rhine-Meuse Delta used in the model. The abbreviations correspond to Fig. 2. The width at the boundaries of the channels are provided here, the width in the interior is computed with Eq. 1.

| Channel | Abbreviation | Depth [m] | Length [km] | Initial width [m] | Final width [m] |
|---|---|---|---|---|---|
| Beneden Merwede | BM | 6 | 15.3 | 100 | 300 |
| Breeddiep | BD | 16 | 2.5 | 1200 | 1200 |
| Dordtsche Kil | DK | 10.7 | 9.4 | 300 | 300 |
| Haringvliet | HV | 8.7 | 11.7 | 2420 | 2420 |
| Hartelkanaal | HK | 7.6 | 25.8 | 310 | 1500 |
| Hollands Diep 1 | HD1 | 7.6 | 32.0 | 1630 | 2000 |
| Hollands Diep 2 | HD2 | 6.2 | 3.81 | 1600 | 1100 |
| Hollandse IJssel | HY | 4.0 | 19.7 | 45 | 150 |
| Lek | LE | 5.3 | 42.0 | 136 | 260 |
| Maas | MA | 5.3 | 62.3 | 97 | 416 |
| Nieuwe Maas 1 | NM1 | 11 | 18.75 | 400 | 500 |
| Nieuwe Maas 2 | NM2 | 8.1 | 4.9 | 250 | 400 |
| Nieuwe Merwede | NM | 5 | 19.6 | 400 | 730 |
| Nieuwe Waterweg | RW | 16 | 16.8 | 600 | 600 |
| Noord | NO | 7 | 8.6 | 250 | 220 |
| Oude Maas 1 | OM1 | 13 | 3.1 | 250 | 250 |
| Oude Maas 2 | OM2 | 14 | 8.25 | 317 | 317 |
| Oude Maas 3 | OM3 | 10.2 | 15 | 240 | 350 |
| Oude Maas 4 | OM4 | 7 | 4.26 | 250 | 250 |
| Spui | SP | 6.4 | 17.4 | 250 | 250 |
| Waal | WA | 4 | 45.2 | 214 | 500 |



## Appendix B: Equations for water motion and salinity

To compute the salt transport $T$ (Eq.3), we need to solve for the horizontal velocity and salinity components in this equation. The derivation of the expressions and equations for these variables are in Biemond et al. (2024a). Analytical expressions for subtidal water level gradient $\frac{\partial \eta_{st}}{\partial x}$, the depth-averaged and depth-varying component of the subtidal horizontal velocity $u_{st}$ and vertical velocity $w_{st}$ are obtained and read

$$\frac{\partial \eta_{st}}{\partial x} = -\frac{6}{5}\frac{A_{v,st}Q}{gbH^3}, \tag{B1a}$$


$$\bar{u}_{st} = \frac{Q}{bH}, \tag{B1b}$$

$$u'_{st} = \frac{Q}{bH}P_1(\tilde{z}) + \frac{g\beta H^3}{48A_{v,st}}\frac{\partial \bar{s}_{st}}{\partial x}P_2(\tilde{z}), \tag{B1c}$$

$$w_{st} = -\frac{g\beta H^4}{48A_{v,st}}\left(\frac{\partial^2 \bar{s}_{st}}{\partial x^2} + L_b^{-1}\frac{\partial \bar{s}_{st}}{\partial x}\right)P_w(\tilde{z}). \tag{B1d}$$

Here, a bar indicates a depth-averaged quantity and a prime a variation from the depth-average, while the subscript $st$ (subtidal) indicates a tidally averaged variable and subscript $ti$ will be used for tidally varying variables. Further, $Q$ is net water transport, $b$

is width, $H$ is depth, $g = 9.81\,\mathrm{m\,s^{-2}}$ is gravitational acceleration, $\beta = 7.6\,10^{-4}\,\mathrm{(g/kg)^{-1}}$ is the isohaline contraction coefficient, $A_{v,st}$ is vertical viscosity acting on the subtidal flow, $s$ is salinity, $x$ is horizontal coordinate and $L_b$ is a length scale that controls the width convergence. Quantity $\frac{\partial \eta_{st}}{\partial x}$ is required to calculate the water distribution over the channels, and $w_{st}$ for the vertical structure of the subtidal salinity field $s'_{st}$. Note that we have neglected the contribution of salinity to the subtidal water level gradient for the same reason as described in Biemond et al. (2023), i.e. taking it into account would introduce a

strong relationship between salinity and net water transport, which is not realistic (Buschman et al., 2010; Maicu et al., 2018; Wang et al., 2021). Polynomials $P_1$, $P_2$ and $P_w$ describe the vertical structure of the flow, and depend on normalised vertical coordinate $\tilde{z} = \frac{z}{H}$. They are defined as

$$P_1(\tilde{z}) = \left(\frac{1}{5} - \frac{3}{5}\tilde{z}^2\right), \quad P_2(\tilde{z}) = \left(\frac{8}{5} - \frac{54}{5}\tilde{z}^2 - 8\tilde{z}^3\right), \quad P_w(\tilde{z}) = \left(2\tilde{z}^4 + \frac{18}{5}\tilde{z}^3 - \frac{8}{5}\tilde{z}\right). \tag{B2}$$

The water level $\eta_{ti}$, horizontal velocity component $u_{ti}$ and vertical velocity component $w_{ti}$ of the tidally varying flow field

read

$$(\eta_{ti}, u_{ti}, w_{ti}) = \Re\{(\hat{\eta}_{ti}, \hat{u}_{ti}, \hat{w}_{ti})\exp(-i\omega t)\}, \tag{B3a}$$

$$\hat{\eta}_{ti} = \exp\left(-\frac{(x+L)}{2L_b}\right)(C_1\exp(k(x+L)) + C_2\exp(-k(x+L))), \tag{B3b}$$

$$\hat{u}_{ti} = \frac{g}{i\omega}\frac{\partial \hat{\eta}_{ti}}{\partial x}\left(B\cosh(\delta_A\frac{z}{H}) - 1\right), \tag{B3c}$$

$$\hat{w}_{ti} = i\omega\hat{\eta}_{ti} - \frac{g}{i\omega}\left(\frac{d^2\hat{\eta}_{ti}}{dx^2} + \frac{1}{L_b}\frac{d\hat{\eta}_{ti}}{dx}\right)\left(\frac{BH}{\delta_A}\sinh(\delta_A\frac{z}{H}) - z\right). \tag{B3d}$$

Here $\omega$ is the angular frequency of the tidal constituent that is considered, $t$ is time, $L$ is the length of a channel segment, $\Re$ denotes the real part of a complex variable and $C_1$ and $C_2$ are determined by the horizontal boundary conditions. The other



parameters are defined as

$$\delta_A = \frac{(1+i)H}{\sqrt{\frac{2A_{v,ti}}{\omega}}}, \tag{B4a}$$

$$B = \left(\cosh(\delta_A) + \frac{\delta_A}{2}\sinh(\delta_A)\right)^{-1}, \tag{B4b}$$

$$k = \sqrt{\frac{1}{(2L_b)^2} + \frac{\omega^2}{gH}\left(\frac{B}{\delta_A\sinh(\delta_A)} - 1\right)^{-1}}, \tag{B4c}$$

in which $A_{v,ti}$ is the vertical viscosity acting on the tidal flow. Note that we split subtidal horizontal velocity in a depth-averaged and a depth-dependent component, but do not make this separation for tidal velocity.

For salinity, no analytical expressions are obtained, but the decomposition from Eq. 2 is substituted in the 2DV mass balance for salinity, which reads

$$\frac{\partial s}{\partial t} + \frac{1}{b}\frac{\partial}{\partial x}(bus) + \frac{\partial}{\partial z}\left(ws\right) = \frac{1}{b}\frac{\partial}{\partial x}\left(bK_h\frac{\partial s}{\partial x}\right) + \frac{\partial}{\partial z}\left(K_v\frac{\partial s}{\partial z}\right). \tag{B5}$$

Here, $t$ is time, and $K_h$ and $K_v$ are the horizontal and vertical eddy diffusivities, respectively. To obtain an equation or subtidal, depth-averaged salinity $\bar{s}_{st}$, we average this equation over depth and the tidal timescale, which gives

$$\frac{\partial \bar{s}_{st}}{\partial t} + \bar{u}_{st}\frac{\partial \bar{s}_{st}}{\partial x} + \overline{u'_{st}\frac{\partial s'_{st}}{\partial x}} + (\bar{u}_{ti}\frac{\partial \bar{s}_{ti}}{\partial x})_{st} + (\overline{u'_{ti}\frac{\partial s'_{ti}}{\partial x}})_{st} = \frac{1}{b}\frac{\partial}{\partial x}(bK_{h,st}\frac{\partial \bar{s}_{st}}{\partial x}). \tag{B6}$$

In this expression, $K_{h,st}$ is the horizontal diffusion coefficient acting on the subtidal salinity. The equation for the vertically varying part of the subtidal salinity $s'_{st}$, obtained from subtracting Eq. B6 from the tidally averaged mass balance, reads

$$\frac{\partial s'_{st}}{\partial t} + \bar{u}_{st}\frac{\partial s'_{st}}{\partial x} + u'_{st}\frac{\partial \bar{s}_{st}}{\partial x} + u'_{st}\frac{\partial s'_{st}}{\partial x} - \overline{u'_{st}\frac{\partial s'_{st}}{\partial x}} + w_{st}\frac{\partial s'_{st}}{\partial z} + (\bar{u}_{ti}\frac{\partial s'_{ti}}{\partial x})_{st} + (u'_{ti}\frac{\partial \bar{s}_{ti}}{\partial x})_{st}$$
$$+ (u'_{ti}\frac{\partial s'_{ti}}{\partial x})_{st} - (\overline{u'_{ti}\frac{\partial s'_{ti}}{\partial x}})_{st} + (w_{ti}\frac{\partial s'_{ti}}{\partial z})_{st} = \frac{1}{b}\frac{\partial}{\partial x}(bK_{h,st}\frac{\partial s'_{st}}{\partial x}) + \frac{\partial}{\partial z}(K_{v,st}\frac{\partial s'_{st}}{\partial z}). \tag{B7}$$

Here, $K_{v,st}$ is the vertical diffusion coefficient, acting on the subtidal salinity. Finally, an equation for tidally varying salinity $s_{ti}$ is constructed, by subtracting the tidally averaged mass balance from the full mass balance. Additionally, only the dominant tidal constituent is considered, i.e. $s_{ti} = \Re[\hat{s}_{ti}\exp(-i\omega t) + c.c.]$, and scaling analysis is applied to select the most important terms. The equation governing the dynamics of $\hat{s}_{ti}$ is

$$-i\omega\hat{s}_{ti} + \hat{u}_{ti}\frac{\partial \bar{s}_{st}}{\partial x} + \hat{w}_{ti}\frac{\partial s'_{st}}{\partial z} = \frac{\partial}{\partial z}(K_{v,ti}\frac{\partial \hat{s}_{ti}}{\partial z}), \tag{B8}$$

in which $K_{v,ti}$ is the vertical diffusion coefficient, acting on the tidally varying salinity. This equation has an analytical solution, i.e. $\hat{s}_{ti}$ can be expressed as a function of $s_{st}$, but this expression is very lengthy and therefore not presented here.

## Appendix C: Boundary conditions at internal boundaries

There are two types of internal boundaries in the model: connections between segments and junctions. The conditions below are given for junctions. For connections between segments the same conditions apply, but only two channels are considered.



The conditions for hydrodynamics at the internal boundaries read

$$\sum_{p=1}^{3} b_p H_p \bar{u}_{st,p} = 0, \quad \eta_{st,1} = \eta_{st,2} = \eta_{st,3}, \quad \sum_{p=1}^{3} b_p H_p \bar{u}_{ti,p} = 0, \quad \eta_{ti,1} = \eta_{ti,2} = \eta_{ti,3}, \tag{C1}$$

where subscripts 1, 2 and 3 are indices of the channels which are connected to the junction. Regarding salinity, continuity of salt transport and salinity is used at every vertical level, which reads

$$\sum_{p=1}^{3} T_p = 0, \quad \bar{s}_{st,1} = \bar{s}_{st,2} = \bar{s}_{st,3}, \tag{C2a}$$

$$\sum_{p=1}^{3} b_p H_p \left( u'_{st,p} s'_{st,p} - \overline{u'_{st,p} s'_{st,p}} + (u'_{ti,p} s'_{ti,p} - \overline{u'_{ti,p} s'_{ti,p}})_{st} - K_{h,st} \frac{\partial s'_{st,p}}{\partial x} \right) = 0, \quad s'_{st,1} = s'_{st,2} = s'_{st,3}, \tag{C2b}$$

$$\sum_{p=1}^{3} b_p H_p \left( u_{ti,p} s_{ti,p} + u_{ti,p} s_{ti,p} - K_{h,ti} \frac{\partial s_{ti,p}}{\partial x} \right) = 0, \quad s_{ti,1} = s_{ti,2} = s_{ti,3}. \tag{C2c}$$

Note that the conditions for the vertically varying quantities are evaluated at sigma levels, i.e. at $\tilde{z} = \frac{z}{H} = $ constant, when the channels have a non-equal depth.

## Appendix D: Boundary layer correction for salinity at junctions

The first-order equation for tidally varying salinity $s_{ti}$ reads (Biemond et al., 2024a)

$$\frac{\partial s_{ti}}{\partial t} + u_{ti} \frac{\partial \bar{s}_{st}}{\partial x} + w_{ti} \frac{\partial s'_{st}}{\partial z} = \frac{\partial}{\partial x}(K_{h,ti} \frac{\partial s_{ti}}{\partial x}) + \frac{\partial}{\partial z}(K_{v,ti} \frac{\partial s_{ti}}{\partial z}), \tag{D1}$$

in which $t$ is time, $u_{ti}$ and $w_{ti}$ are the horizontal and vertical component of the tidal current, respectively, $\bar{s}_{st}$ is the depth and tidally averaged salinity, $s'_{st}$ is the vertical distortion from the tidally-averaged salinity and $K_{h,ti}$ and $K_{v,ti}$ are horizontal and vertical diffusivity, acting on the tidal salinity. The boundary conditions are no flux at the bottom and surface, and condition Eq. C2c for the horizontal boundaries (junctions). The horizontal eddy diffusion term is included in Eq. D1. In Wei et al. (2016) and Biemond et al. (2024a) it was argued that this term is small with respect to the other terms in this equation. We argue here that horizontal eddy diffusion is important in the neighbourhood of junctions, as tidal currents are observed to be more turbulent close to junctions (Corlett and Geyer, 2020), which increases the magnitude of eddy diffusion. Moreover, when horizontal diffusion is excluded, Eq. D1 does not contain horizontal derivatives to $s_{ti}$, which implies that no horizontal boundary conditions can be applied, and therefore continuity of salinity at the junctions can not be imposed.

To develop a set of equations which takes horizontal eddy diffusion into account around the junctions, we scale Eq. D1 by defining

$$t = \omega^{-1} \tilde{t}, \ x = L\tilde{x}, \ z = H\tilde{z}, \ u_{ti} = U\tilde{u}_{ti}, \ w_{ti} = H\omega \, \tilde{w}_{ti}, \ s_{st} = S_{st} \tilde{s}_{st}, \ s_{ti} = \frac{U S_{st}}{\omega L} \tilde{s}_{ti}. \tag{D2}$$

In these expressions, $\omega$ is the frequency of the tide, $L$ the horizontal length scale on which $s_{ti}$ varies (when not being affected by junctions), $U$ the magnitude of the tidal current, and $S_{st}$ is a typical scale for the subtidal salinity. The last condition follows





from $\frac{\partial s_{ti}}{\partial t} \approx u_{ti} \frac{\partial s_{st}}{\partial x}$ (Biemond et al., 2024a). This yields, after dropping the tildes,

$$\frac{\partial s_{ti}}{\partial t} + u_{ti} \frac{\partial s_{st}}{\partial x} + \chi_1 w_{ti} \frac{\partial s_{st}}{\partial z} = \varepsilon \frac{\partial^2 s_{ti}}{\partial x^2} + \chi_2 \frac{\partial^2 s_{ti}}{\partial z^2},$$ (D3)

with $\chi_1 = \frac{ZL\omega}{UH}$, $\chi_2 = \frac{K_v}{\omega H^2}$ and $\varepsilon = \frac{K_h}{\omega L^2}$. We write tidally varying salinity as $s_{ti} = \Re[\hat{s}_{ti} \exp(-i\omega t) + c.c.]$ (see Appendix B) and find the equation

$$\chi_2 \frac{\partial^2 \hat{s}_{ti}}{\partial z^2} + i\hat{s}_{ti} + \varepsilon \frac{\partial^2 \hat{s}_{ti}}{\partial x^2} = \hat{u}_{ti} \frac{\partial s_{st}}{\partial x} + \chi_1 \hat{w}_{ti} \frac{\partial s_{st}}{\partial z}.$$ (D4)

When choosing typical scales for an estuary ($H = 10$ m, $L = 10$ km, $U = 1$ m s$^{-1}$, $K_{h,ti} = 25$ m s$^{-2}$, $K_{v,ti} = 0.01$ m s$^{-2}$ and $\omega = 1.4 \cdot 10^{-4}$ s$^{-1}$), it is found that $\chi_1$ and $\chi_2$ are of order one and $\varepsilon$ is a factor $10^3$ smaller. This defines a singular perturbation problem, which is solved using standard techniques (see e.g. Eckhaus (2011)).

To construct the regular solution, we write

$$\hat{s}_{ti,reg} = \sum_{n=0}^{\infty} \varepsilon^n \phi_n.$$ (D5)

Inserting this in Eq. D4 and collecting the first order terms gives Eq. B8, as expected, which is valid outside the neighbourhood of the junctions. Close to the junctions, we look for a correction to this equation, resulting from the horizontal diffusion. We write $\psi = \hat{s}_{ti} - \hat{s}_{ti,reg}$, and substitute this, which yields

$$\chi_2 \frac{\partial^2 \psi}{\partial z^2} + i\psi + \varepsilon \frac{\partial^2 \psi}{\partial x^2} = -\chi_2 \frac{\partial^2 \hat{s}_{ti,reg}}{\partial z^2} - i\hat{s}_{ti,reg} - \varepsilon \frac{\partial^2 \hat{s}_{ti,reg}}{\partial x^2} + u_{ti} \frac{\partial s_{st}}{\partial x} + \chi_1 w_{ti} \frac{\partial s_{st}}{\partial z}.$$ (D6)

We perform an expansion for the perturbation, i.e. $\psi = \sum_{n=0}^{\infty} \varepsilon^n \psi_n$. The first order solution then reads

$$\chi_2 \frac{\partial^2 \psi_0}{\partial z^2} + i\psi_0 + \varepsilon \frac{\partial^2 \psi_0}{\partial x^2} = 0.$$ (D7)

Now a boundary layer coordinate $\zeta$ is introduced, i.e. we write $\psi_0 = B(z)e^{\zeta}$, with $\zeta = \pm \frac{x}{\varepsilon^\nu}$. It follows that $\nu = \frac{1}{2}$ when the horizontal diffusion term is of equal magnitude as the other terms in Eq. D7. Substitution of this gives the following equation for $B(z)$:

$$\chi_2 \frac{d^2 B}{dz^2} + iB + B = 0.$$ (D8)

The no-flux boundary conditions at the bottom and surface apply to this equation. This is a homogenous problem, which has solution

$$B(z) = \sum_{m=0}^{\infty} B_m \cos(m\pi \frac{z}{H}).$$ (D9)

The values of $B_m$ follow from the horizontal boundary conditions associated with the junction (Eq. C2c). With this, we have a solution for tidally varying salinity in the neighbourhood of the junctions (the extent of this region is determined by

$\varepsilon^{\frac{1}{2}}$), which accounts for increased eddy diffusion around the junctions and obeys the horizontal boundary conditions. We set $K_{h,ti} = 25$ m$^2$s$^{-1}$; sensitivity tests indicate that the model skill is insensitive to this value.




## Appendix E: Derivation of the dependence of salt intrusion on the sign of the change in depth

To explain why an increase in depth of the Rotterdam Waterway (RW) has a smaller effect on salt intrusion in the Nieuwe and Oude Maas than a decrease, we develop a set of equations to study the difference between changing the depth of the downstream part of a channel versus the depth of the entire channel.

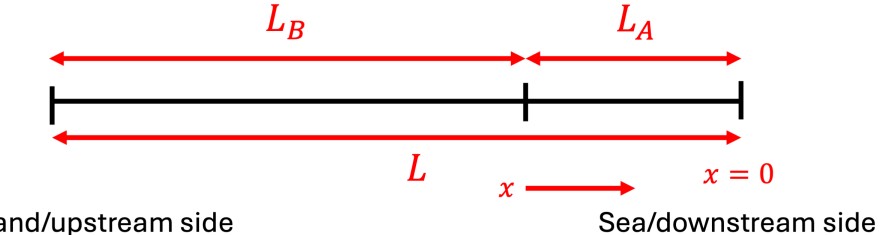

**Figure E1.** A sketch of the domain used in Appendix E.

Suppose we have a single-channel estuary with constant width in equilibrium, where the salt import is governed by exchange flow, like in the RW. In that case, the terms $T_Q$ and $T_E$ in Eq. 3 make balance with each other. If we furthermore assume that stratification is the result of a balance between advection of $\bar{s}_{st}$ by the vertical shear of the subtidal current and vertical diffusion (Pritchard, 1954), the following equation for depth-averaged subtidal salinity $\bar{s}_{st}$ holds (Chatwin, 1976):

$$\frac{Q}{bH}\bar{s}_{st} = \frac{0.112}{48^2}\frac{g^2\beta^2 H^8}{A_{v,st}^2 K_{v,st}}\left(\frac{\partial\bar{s}_{st}}{\partial x}\right)^3. \tag{E1}$$

The domain is shown in Fig. E1 and the associated boundary condition is $\bar{s}_{st} = s_{oc}$ at $x = 0$.

In a single-channel estuary with length $L$ and constant width, this equation has an analytical solution for salinity, which reads

$$\bar{s}_{st} = s_{sc}\left(\left(\frac{s_{oc}}{s_{sc}}\right)^{\frac{2}{3}} + \frac{2}{3}\frac{x}{L_e}\right)^{\frac{3}{2}} \quad \text{with} \quad L_e = \left(\frac{0.112}{48^2}\frac{bH}{Q}\frac{g^2\beta^2 s_{sc}^2 H^8}{A_{v,st}^2 K_{v,st}}\right)^{\frac{1}{3}}, \tag{E2}$$

where we have introduced salinity scale $s_{sc}$. To be able to change the depth of the downstream part of the channel separately, the single-channel estuary is divided in two segments with lengths $L_A$ and $L_B$ (see Fig. E1). In these segments, Eq. E2 with $L_e \rightarrow L_{e,A}$, the value of $L_e$ in domain A, is valid for segment A, and for segment B we have

$$\bar{s}_{st} = s_{sc}\left(\left(\frac{s_{oc}}{s_{sc}}\right)^{\frac{2}{3}} - \frac{2}{3}\frac{L_A}{L_{e,A}} + \frac{2}{3}\frac{(x+L_A)}{L_{e,B}}\right)^{\frac{3}{2}}, \tag{E3}$$

where we have used continuity of salinity at the segment boundary (see Eq. C2a). Note that the values for $L_{e,A}$ and $L_{e,B}$ are different when the depths of the segments differ. From these expressions, we can find the position of a certain isohaline $X_2$ where salinity equals $s_{tres} = 2$ g kg$^{-1}$. This yields for a one segment estuary

$$X_2 = \frac{3}{2}L_e\left(\left(\frac{s_{oc}}{s_{sc}}\right)^{\frac{2}{3}} - \left(\frac{s_{tres}}{s_{sc}}\right)^{\frac{2}{3}}\right). \tag{E4}$$



For an estuary consisting of two segments, Eq. E4 (with $L_e \to L_{e,A}$) is also valid if the salt intrusion is confined to the first segment. In case $X_2$ is located in segment B, it follows

$$X_2 = L_A + \frac{3}{2}L_{e,B}\left(\left(\frac{s_{oc}}{s_{sc}}\right)^{\frac{2}{3}} - \left(\frac{s_{tres}}{s_{sc}}\right)^{\frac{2}{3}} - \frac{2}{3}\frac{L_A}{L_{e,A}}\right). \qquad (E5)$$

These expressions allow for studying the effect of changes to the depth of the entire channel and for changes to the depth of only the downstream segment.

*Author contributions.* Bouke Biemond: Writing – original draft, Visualization, Validation, Software, Methodology, Investigation, Formal analysis. Wouter Kranenburg: Writing – review & editing. Ymkje Huismans: Writing – review & editing. Huib E. de Swart: Writing – review
& editing, Supervision, Funding acquisition. Henk A. Dijkstra: Writing – review & editing, Supervision, Project administration, Funding acquisition. Authors Wouter Kranenburg and Ymkje Huismans contributed equally to this study.

*Competing interests.* The authors declare that they have no competing interests which are relevant to this study.

*Acknowledgements.* This work is part of the Perspectief Program SaltiSolutions, which is financed by NWO Domain Applied and Engineering Sciences (2022/TTW/01344701 P18-32 project5) in collaboration with private and public partners.



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
