# Peer review of "Dynamics of salt intrusion in complex estuarine networks; an idealised model applied to the Rhine-Meuse Delta"

_EGUsphere, 2024_

## Author Response (AR1)

**Comments reviewer 1**

The manuscript by Biemond et al. investigates salt intrusion in complex estuarine networks, focusing on the Rhine-Meuse Delta as a prototype example. The authors develop and apply an idealized 2D model to understand the mechanisms contributing to salt fluxes. I believe this reduced-order model is a promising approach for understanding salt fluxes in multiple channel systems. The results provided by the model regarding the Rhine-Meuse Delta also offer valuable insights into the understanding of estuarine dynamics. My primary concern lies with the interpretation of the tidal dispersion term in the model. I believe the manuscript could be publishable after revisions to address this concern.

We thank the reviewer for the careful reading and useful comments. In our reply below, the line numbers refer to the revised version of the manuscript.

Major comments:

The term $T_D$ is referred to as both tidal dispersion and diffusion throughout the manuscript. It would be helpful to clarify the terminology. My understanding is that diffusion typically relates to turbulent diffusion and molecular diffusion, which are not necessarily isotropic but generally have similar magnitudes in the vertical and horizontal directions. In your case, the horizontal dispersion/diffusion coefficient $K_{h,st}$ = 275 m^2/s is quite large, so I would call it a dispersion coefficient (resulting from the interaction of diffusion and advection processes rather than just diffusion).

We agree that the term 'dispersion' is more appropriate than 'diffusion'. We therefore replaced 'diffusion' with 'dispersion' where appropriate (i.e., when it concerns horizontal dispersion).

I am particularly interested in the $T_T$ term, the time correlation between velocity and salinity, although it seems not to be the main focus of the manuscript. This term is sometimes referred to as oscillatory tidal dispersion and is a dominant term in many shallow estuarine channels. Therefore, it is important to ensure that $T_D$ and $T_T$ do not overlap in their definitions in your model, as they can both be referred to as forms of dispersion. Please clarify this in the manuscript.

In our model formulation, $T_T$ is the resolved part of the tidal salt transport, and $T_D$ is the unresolved part of the total upstream salt transport. To clarify the interpretation of these terms, we added after line 159: 'In our model, unresolved upstream salt transport processes are parametrised by the latter.'

(Biemond et al. (2024) presents a more detailed analysis of the properties of $T_T$ in this model.)

My concern is that the model appears limited in its ability to resolve the $T_T$ term. In the model, exchange at network junctions is pretty much the only mechanism that can lead to $T_T$ (I think the vertical advection of subtidal salinity that you mentioned in the manuscript is negligible in most estuaries, so its contribution to $T_T$ should not be important).

Note that in Biemond et al. (2024) it is found that the latter mechanism explains a substantial part of the total salt transport in the Guadalquivir and Delaware estuaries.

However, there are other mechanisms that can contribute to $T_T$, but not represented here, for example, jet-sink exchange at the river mouth, also referred to as tidal pumping (e.g., Chen et al., 2012). This mechanism depends on variations in horizontal flow structures, which cannot be resolved in the 2D width-averaged model. Also, other complex 3D topographic features can lead to oscillatory tidal dispersion (Dronkers and van de Kreeke, 1986; Garcia & Geyer, 2023). I am concerned that because some mechanisms contributing to $T_T$ are not represented in the 2D model, their associated salt fluxes may be inaccurately captured within the $T_D$ term when tuning the model.

In addition, it seems that only a single $K_{h,st}$ is used for all the channel. In reality, $K_{h,st}$ is likely to vary significantly between different channels due to varying dispersion mechanisms. Additionally, $K_{h,st}$ is determined based on a specific set of observations, but it could change with varying forcing conditions like river discharge and with changing channel depth. This is a limitation as you discuss the responses to different forcing conditions by using a constant $K_{h,st}$.

Indeed, the assumption of a 2DV model implies that certain processes (like the ones mentioned by the referee) are not resolved. And indeed, assuming a constant horizontal dispersion coefficient is a gross simplification of reality. There are also several other strong simplifications made (e.g. constant vertical eddy viscosity and eddy diffusivity, constant depths of the channels, neglecting water intakes). These model limitations are now specifically addressed in a new section in the discussion (Section 4.4) and suggestions for relaxing them are given.

One should keep in mind that this work is an exploratory study, which aims at studying the salt intrusion dynamics in estuarine networks and does not focus on the tidal salt transport itself. We now better explain this model approach in a new Section 2.1.

Overall, the manuscript contains sufficient publishable material. It is understandable that the concerns raised may not be fully addressed within the framework of a 2D model. However, I think the manuscript would benefit from a more extensive discussion of the model's limitations, based on the concerns raised above:

(1) The relationship between terms T_T and T_D.

(2) The use of a single K_H,st value for all the channels.

(3) The potential variability of K_H,st with difference forcing conditions.

We addressed these points following our response above.

Here are some references mentioned in the comments above:

Chen, S. N., Geyer, W. RS), e2022JC018883.

Minor comments:

1. Equation (2): What is the role of vertical velocity in your model? I am curious if it can significantly affect salt flux or it is mostly negligible.

Fig. 2 in Biemond et al. (2022) shows that subtidal vertical velocity (in a model of similar complexity) is important under high discharge conditions for vertical salt transport, and therefore in determining the stratification. Eq. 14 in Biemond et al. (2024) shows that tidally varying vertical velocity is important to the tidal salt transport. We mention in line 114 in the revised manuscript that 'w_st is required for the vertical structure of the subtidal salinity field'. Also, following suggestions of other reviewers, the revised manuscript includes a more extensive model description, which more explicitly shows the role of vertical velocity in our model.

2. Equation (3): K_h,st is introduced here, but is is not explained until Line164. Consider adding a brief explanation to it immediately following this equation.

In our revised manuscript, we included more model equations in the main text (before this equation), which gives context for the interpretation of K_h,st.

3. Equation (4): Are boundary conditions like u_ti=0 and s_ti = 0 needed? Or could they make your equations overdetermined? And should s_riv be zero?

We rephrased the text such that the boundary condition for tidal motion is more clearly described. There is no boundary condition for s_ti, because its governing equation has no horizontal derivatives (except around the junctions as described in Appendix D). Regarding s_riv, the value of salinity in the rivers is generally non-zero. We use a constant value of 0.15 psu, which is appropriate for the Rhine (see e.g. Fig 4d).
To clarify this in the text, we will rewrite lines 166-171 as:

"Regarding the boundary conditions at river boundaries: subtidal discharge is prescribed, tides are assumed to dissipate in the river beyond the boundary (without reflection at a horizontal boundary), and subtidal salinity is set to the river salinity. Hence

bH  ust = Qriv; C1 = 0;   sst = sriv; s'st = 0: (12)

Here, Qriv and sriv are the river discharge and salinity, respectively, which are generally non-zero. The condition C1 = 0 physically means that there is no downstream travelling tidal wave."

4. L118-121. What is meant by "away from the weir" and "toward the weir"? I would guess that it is landward when the discharge is away from the weir, and seaward when the discharge is toward the weir. If so, the boundary conditions seem reversed. I think seaward discharge should be based on a prescribed salinity, and landward discharge should be based on calculated salinity.

We meant with 'toward the weir' with what the reviewer indicates as landward. Note that landward and seaward also are ambiguous terms, as for instance for the Haringvliet sluices the discharge is seaward when the discharge is towards the weir.

To clarify this, we rephrased "away from the weir" and "toward the weir" as 'at the weir into the domain' and 'at the weir out of the domain'.

5. "The water level at this boundary is chosen in such a way that at the estuary mouth the imposed tidal water level is reproduced." This sounds like it requires quite some manual input. Given that factors such as geometry, friction, and tidal frequency can all affect how the water level propagates from sea boundary to the mouth. It would be helpful to include more detail on how this process is managed.

Since the tidal currents and water level are explicit functions in the model, a matrix equation can be solved for the coefficients in the expression for water level. We explained this in the revised version by rewriting line 183-187 as: "Regarding conditions at sea boundaries (estuary mouths), water levels \eta_st and \eta_ti are prescribed here. To obtain conditions for salinity, additional segments are added that extend seaward from the

mouths. These segments are characterised by strongly increasing widths away from the mouths. At the outer sea boundaries of these segments, salinity is set to the sea salinity. Furthermore, assuming that at these locations there is only an incoming tidal wave (i.e. travelling from sea to estuary), allows us to compute the water levels and tidal flow in these segments. The assumption made here is reasonable, because of the strongly increasing width of these segments."

6. Figure 5. What are the black lines (T_O)?

This is the sum of all the salt transport components. We realized that this notation is not clear. Therefore, we refer to them in the revised version just as 'T', the total salt transport. To this end, we removed the phrase 'The net transport through a channel in equilibrium is called the salt overspill T_o.' Furthermore, in lines 447-450 we replaced T_o with T.

7. Figure 7. The results in this figure are very interesting, but it took me quite a while to understand.

What does \Delta Q represent? Initially, I thought \Delta Q>0 indicated a high discharge event (wet) and \Delta Q<0 is a low discharge event (dry). But given that \Delta Q only has positive values, I was once confused how that represents both wet and dry conditions.

Also, \Delta S>0 for all the cases. But I would guess that salinity increases (\Delta S>0) with a decrease in discharge and decreases (\Delta S<0) with an increase in discharge.

Including more explanations on \Delta Q and \Delta S in the figure caption and main text would help readers understand it better. I hope this feedback is useful as you make these changes.

We chose to use absolute values of changes in discharge \Delta Q and changes in salt content \Delta S, because the imposed absolute changes in discharge are equal for a decrease or an increase in discharge, and the same holds for the salt content. To make this more clear, we added in the caption that \Delta Q and \Delta S refer to the absolute value of changes in discharge and salt content.

8. L348-383: These two paragraphs are lengthy and difficult to follow, particularly because they describe many specific locations. I suggest breaking them into shorter paragraphs to enhance readability.

We split the text in the results and discussion section into shorter paragraphs to improve readability.

9. L389-404. I am glad to see the discussion on the single $K_{H,st}$ value here. So maybe expand on this discussion to include the other concerns raised above. Remember to split the text into shorter paragraphs for better readability.

This discussion is extended as described in our response to the main concerns raised above. Additionally, we split the discussion section into shorter paragraphs.

**References**

Biemond, B., de Swart, H. E., Dijkstra, H. A., & Díez-Minguito, M. (2022). Estuarine salinity response to freshwater pulses. *Journal of Geophysical Research: Oceans*, 127, e2022JC018669. https://doi.org/10.1029/2022JC018669

Biemond, B., de Swart, H. E. & Dijkstra, H. A. (2024). Quantification of salt transports due to exchange flow and tidal flow in estuaries. Journal of Geophysical Research: Oceans, accepted for publication. Doi: doi.org/10.48550/arXiv.2408.06378

**Comments reviewer 2**

Review of: Dynamics of salt intrusion in complex estuarine networks; an idealised model applied to the Rhine-Meuse Delta

Author(s): Bouke Biemond et al.

MS No.: egusphere-2024-2322

https://doi.org/10.5194/egusphere-2024-2322

This paper describes the development of a reduced physics numerical modeling system to simulate the salt transport through a network of tidal channels. Results are analyzed to explain different mechanisms responsible for salt transport. Sensitivity tests are performed to evaluate changes in fresh water discharge and bathymetry. In general the paper is well organized and explains the approach clearly. Results are clearly explained. My main concern is that the model only explains ~67% of the salinity variance, and the fundamental processes may be missing. Also, the mixing coefficients needed to be calibrated, but when there are dramatic changes to the system (Qriver or bathy), those coefficients may need to be modified.

We thank the reviewer for the careful reading and useful comments. In our reply below, the line numbers refer to the revised version of the manuscript.

The reviewer is correct that not all variance contained in the observations is resolved in the model and that not all physical processes are explicitly resolved. However, the model approach is that of an exploratory model, in which comparison with observations serves as a way to get the values of the model parameters in the appropriate range, but getting an as good as possible correspondence with the observations is not a primary goal. To make this clearer, we added a new section (2.1) before the model description, in which we describe the modelling approach.

Next, we agree with the reviewer that in principle the mixing coefficients will change when the bathymetry of the system changes. We introduced a new section (4.4) to the discussion, also in line with comments from the first reviewer, to stress that a more detailed analysis of changes in bathymetry can be performed when a more sophisticated formulation of the mixing coefficients is used.

Here are some specific comments below.

1) Lines 49-50 state "This model is suitable for a process-based analysis of salt intrusion mechanisms, ..." What are some of the disadvantages of your approach?

Disadvantages in this aspect are that processes are neglected based on scaling arguments, which limit the parameter space the model is applicable to. We added a new section to the discussion (Section 4.4) in which we discuss these disadvantages, but also present suggestions of how these disadvantages could be reduced.

2) Line 51 "and is more flexible in terms of e.g. estuarine geometry" But those unstructured grid models can do the geometry rather well.

We meant here that the construction of the geometry itself is simple. Construction of a suitable unstructured grid for a multi-channel system like the Rhine-Meuse Delta (RMD) is not a simple task and depends on availability of detailed bathymetry data. However, the construction of the geometry files of our model for a system like the RMD takes little effort, and the geometry of other systems is equally simple to implement (salinity data to validate the model is more of a concern). To make this clearer, we modified line 50-51 as ' Furthermore, it has lower computational costs than a 3D numerical model and it is e.g. easier to implement estuarine network geometry.'

3) line 77: How many segments in each channel length? Were the results sensitive to the number of segments?

The number of segments is one or two, as mentioned in line 195. Multiple segments could be used when the geometry of a channel varies strongly. It turned out that this was not essential to model the RMD and therefore we do not use this option here. Additionally, more than one segment is used if we want to use a different numerical grid size in parts of the channel (and for the channels which connect to the sea or rivers), but this has no effect on the physics of the model. For instance, the Spui is separated into two segments, of which the downstream segment has dx = 370 m (salinity can reach this part) and the upstream segment has dx = 1000 m (salinity rarely reaches this part). To clarify this, we modified lines 195-196 as ' The channels consist of one or two segments. The division in multiple segments is based on the desired horizontal grid size and does not affect the physics.'

4) line 96: How can the tidal current uti not depend on the salinity?

The Ianniello (1979) solution of tidal currents in an estuary, which we use here as well, assumes that vertical eddy viscosity and bottom slip parameter (their components that act on the tidal flow) do not depend on salinity. This is mentioned in the revised version in line 131-132: 'Furthermore, $A_{v,ti}$ and therefore $u_{ti}$ do not depend on salinity.'

5) lines 107-108 "The values of vertical viscosity and diffusivity and horizontal diffusivity are assumed to be constant throughout the entire domain." This is a strong limitation.

This indeed limits the salt dynamics in the model. We added in the discussion (Section 4.4) that this is an interesting topic for future research.

6) line 146: Why is only the M2 imposed? How representative of the tide is this constituent?

Again, to focus on the network behavior, we kept the tides simple and only considered the dominant constituent, which is the M2 component in the Rhine-Meuse Delta. We mentioned the quantitative importance of this component in the revised version in lines 204-208: "The model is forced with prescribed time series of discharge at the two river and the three weir boundaries, and with the water level amplitude and phase of the dominant tidal constituent, which is the semi-diurnal lunar M$_2$ tide (period 12 h, 25 m) for the RMD, at the North Sea boundary (note that the water level amplitude of the M$_2$ component at the station at the mouth (b$_1$) is four times larger than that of the second largest component (Walters, 1987))."

7) line 186 says "The overall Nash-Sutcliffe efficiency (Eq. 7) is 0.67, which classifies the model performance as satisfactory (Moriasi et al., 2015)." This is still rather low, and the overall mechanisms for salt transport may not be adequately resolved.

The reasons for this difference are given in lines 253-255. As mentioned above, our view is that it is encouraging that, in light of the strong simplifications (constant viscosity and mixing coefficients, constant depth of channels, no wind forcing), the main observed salinity patterns are resolved. In the revised version, we added a section at the beginning of the model description (2.1) which describes the model philosophy, such that we better explain that an as good as possible model-observation is not a goal of this study. We also provide suggestions for model improvements (Section 4.4), which will likely result in higher values of the NSE.

8) lines 187-188: text reads "This is due to the fact that our model does not account for the spring-neap tidal cycle, overtides and subtidal and intratidal water level fluctuations at the sea boundary driven by remote winds, which have a strong influence on the variability of

the salinity." It is good to see that you are specifying the reasons for the underestimation. This could be stated earlier as a limitation. Why can't these lower frequencies be included on the open boundary?

In principle, they can be included. This is an interesting direction of future research, but beyond the scope of this study. We suggest this now in the revised discussion, in subsection 4.4: "We present here recommendations for model aspects which could be improved in further work ... The most promising directions to achieve this are inclusion of additional mechanisms which generate a phase difference between tidal currents and salinity, e.g. multiple tidal constituents (externally forced and internally generated) ... Another direction for future research is the inclusion of additional forcing conditions, e.g. water level fluctuations at subtidal and intratidal time scales at the sea boundary (Kranenburg et al., 2022)"

9) line 192: The text is referring to figs S1d vs S1f. But station a1 is shown on figs S1a and S1c.

Thanks for pointing this out. In the revised version, we refer to Fig. S1b and S1d (note that the indexing is changed because a panel is added to address the next point).

10) Looking at the Supplemental time series fig S1, the main pulse of salinity that entered the domain at day ~210 seems underestimated from a1 (surface mostly) to a2 to a3. The trib a4 gets the peak but the trib to a5 is too low. What causes that sharp peak of salinity?

To make the relation between salinity and applied forcing clearer, we added a panel to Fig. S1, which contains the total discharge into the delta as a function of time. The particular event around day 210 turns out to be related to a drop in discharge.

11) Can you show where the Waal and Maas rivers, Haringvliet sluices, Hollandse IJssel and Lek are?

We added the location of these boundaries to Fig. 1.

12) Does a time series of Q river compare inversely to the salinity time series in Fig 4 or the S1?

This relationship between discharge and salinity holds for single-channel estuaries. However, in estuarine networks, this relationship is more complex, as different channels have different values of discharge, and salt fluxes of connecting channels influence each other. As an example, we show a plot of total Q vs s at point $a_6$ below, which shows that s decreases with Q, but quite some scatter is present, due to the e.g. network effects and

time lags. To properly entangle the different effects which generate the scatter requires some dedicated analysis and is beyond the scope of this study.

[Figure]

*Figure 1 Subtidal salinity at point $a_6$ (z=-9.0m) as a function of discharge for the simulation of the year 2022.*

13) Figure S2- why is HK so well mixed?

Because this channel is much shallower than its neighboring channels (see Table A1: HK is 7.6 m deep and e.g. RW is 16 m deep). We added this fact to lines 315-317: ' A similar mechanism plays a role in the HK, which is shallower than its neighbouring channels, therefore receives less discharge (71 m3s−1, about 9 times smaller than the RW) and consequently is less stratified and the channel-averaged contribution of T_E to the upstream salt transport in the HK is only 4% (Fig. 5d).'

14) What is To in fig 5? Total?

This is indeed the total salt transport. We realised that our notation created confusion and therefore just refer to this transport as 'T' in the revised version.

15) line 217- do you think the vertical advection is so important because you used a constant eddy viscosity?

This is an interesting question and one of the topics for future studies, as is now identified in Section 4.4.

16) Figure 6 – This phase lag and trapping effect would be subjective to the along channel grid spacing.

The horizontal grid spacing indeed should be sufficient to resolve the boundary layer around the junctions. Tests with the model (not shown) have indicated that when the grid resolution is insufficient around the junctions, wiggles are present in the salinity field around the junctions. No changes in the text required.

17) line 269 – can you use the 90% because the system will reach an equilibrium with the changed steady state conditions?

Indeed, if one is working with data with a high variability, this approach will not work. We made this clear by adding after line 342: '(Note that S goes to a constant value after the change in discharge)'.

18) It is a little surprising that twet is not sensitive to Q river scaling. I would assume that smaller changes in Qriver would take longer to reach an equilibrium. Is this saying that the whole system responds in basically 2.5 days for any increase in discharge (except for HK and HY)?

Correct. The reasons for these dynamics are explained in Biemond et al. (2022) and repeated in lines 354-359.

Does this system have a low residence time?

The fact that the change in tidally averaged salt content of a channel to a change in discharge is small does not necessarily imply that the residence time of the system is low. For instance, a channel may exchange a lot of water on tidal timescales with the adjacent sea, but this does not affect its response time of salinity to changes in discharge. We therefore do not mention residence time in our manuscript.

19) When you alter the flows and depths, there would probably be a need to recalibrate the system for new Kv etc values. Do you think the mixing coefs are still the correct?

The coefficients are calibrated for the current geometry of the delta and changes are expected when the geometry changes. This would be a good topic for a follow-up study (including accounting for spatial variability of these coefficients). Note that the calibration holds for changes in discharge, as various discharges occur during the calibration. In the revised discussion we added that (lines 504-506) "Also, the vertical eddy viscosity, vertical eddy diffusivity and horizontal dispersion coefficients are assumed to be constant in space and time in our model. Considering these coefficients to vary with bathymetry, would lead

to a different response to changes in bathymetry." We now also discuss in Section 4.4 how to relax these conditions.

20) Are there locations in the system where the tide is propagating from both sides, such as in the HK, and there is a null point in the middle? The phase locations can move based on the S/N tidal cycle.

There are no amphidromic points in the RMD as is visible in Fig 3a.

21) How well are the tidal currents simulated?

We did not compare our model data with observations of tidal currents, because there are only a few measurement stations available in the Rhine-Meuse Delta (see https://waterinfo.rws.nl/#/expert/Stroming?parameters=Stroomsnelheid___20Oppervlakt ewater___20m___2Fs). However, the water levels are well represented (see Fig. 3) and therefore we believe that the tidal currents are also satisfactorily resolved. We mention in the revised version in line 216 that: 'There are only a few permanent current measurement stations available in the RMD and therefore currents are not evaluated.'

**References**

Ianniello, J. P. (1979). Tidally induced residual currents in estuaries of variable breadth and depth. Journal of Physical Oceanography, 9 (5), 962–974. doi:10.1175/1520-0485(1979)009⟨0962:TIRCIE⟩2.0.CO;2

Biemond, B., de Swart, H. E., Dijkstra, H. A., & Díez-Minguito, M. (2022). Estuarine salinity response to freshwater pulses. *Journal of Geophysical Research: Oceans*, 127, e2022JC018669. https://doi.org/10.1029/2022JC018669

**Comments editor**

Decision to come once the interactive discussion closes (it's afternoon of Oct 11 here for me already). I have a few writing comments.

We thank the editor for the careful reading and useful comments. In our reply below, the line numbers refer to the revised version of the manuscript.

* general: I get why a lot of the model and mathematical details are dumped into the appendix, but I personally think the authors have gone too far the other way. There are instances where the equations or at least the outline/summary of key results should in my opinion be presented in the main text, even if most of the derivation is left in the appendix; to me it makes the reading abrupt as the reader has to then make a big jump, which disturbs the flow of the reading unnecessarily. Specific cases mentioned below. (I think the material in the appendix is actually the most interesting, but that's my personal bias.)

We adjusted the text following the specific points mentioned below.

line 34 and 36: Would put the reference at the end of the sentence.

We followed this suggestion.

line 44: Suggest "On the other hand, the 3D models..., and while these models typically...affect the salt transport. Their high computational costs makes extensive sensitivity..."

We followed this suggestion.

sec 2.2: I would like the authors to include/recap/outline the relevant equations here for self-containment purposes.

In the revised version, we included the analytical expressions for the velocities and the governing equation for the salinity dynamics in this section.

line 97: "...in the horizontal, and a backward Euler scheme for time integration is employed." (Are the relevant equations linear? If so then no iterative solver is required, but the reader wouldn't no particularly since the relevant equations were not shown.)

We followed the suggestion for the rephrasing. The equations are non-linear, since subtidal velocity depends on salinity. Therefore, an iterative solver is required.

line 107: "The values of vertical viscosity, vertical diffusivity and horizontal diffusivity..."

We followed this suggestion.

line 108: In line with other referees' comments, is it realistic to have constant diffusivity? How sensitive are the associated results and conclusions? Please comment in the paper accordingly.

Following feedback of other reviewers, we rewrote part of the discussion. The revised version mentions more explicitly the implications of assuming constant coefficients and relaxing this assumption is mentioned as an important topic for further study.

line 109-110: It's "different" sure, but what is exactly being done?

To clarify this, we rewrote this sentence as: 'Note that the values of the coefficients used for viscosity, diffusivity and friction depend on whether they act on the tidal or subtidal current and salinity, as is explained in e.g. Godin (1991, 1999)'.

line 111: Clarifications on why tides are assumed to vanish. Looking for an equilibrium solution? If so, please say so.

The tides do not vanish at the river boundary, but are let through without interaction, and are assumed to decay in the river, such that there is no reflection on the river boundary. This text is rewritten as:

"Regarding the boundary conditions at river boundaries: subtidal discharge is prescribed, tides are assumed to dissipate in the river beyond the boundary (without reflection at a horizontal boundary), and subtidal salinity is set to the river salinity. Hence

$b H \bar u_{st} = Q_{riv}$, $\bar s_{st} = s_{riv}$, $s'_{st} = 0$.

Here, $Q_{riv}$ and $s_{riv}$ are the river discharge and salinity, respectively, which are generally non-zero.

Additionally, the condition that there is no downstream travelling tidal wave is prescribed by setting either $C_1$ or $C_2$ to zero, depending on the sign of the real and imaginary parts of $k$."

eline 117: Not clear what "latter" refers to here.

We rephrased this as 'At weir boundaries we prescribe subtidal discharge and use a reflecting boundary condition for the tidal flow, so the tidal flow vanishes at such boundaries.'

line 122: "This reads" -> "In summary, the boundary conditions reads" (or similar)

We followed this suggestion.

1st paragraph of 2.3: For consistency, write out not figure numbers in full (e.g. "21" -> "twenty-one"), because this is done later (e.g. next paragraph, "two river and the three weir boundaries")

We followed this suggestion.

line 141: Probably "...details (e.g. the harbour basins) are..."

We followed this suggestion.

line 143: "The horizontal grid size is around a few hundred meters in general, but differs..." (as written it is oddly precise as implied by the "is", but also vague with the "few")

We followed this suggestion.

line 153: "19" and "7" written out in full

We followed this suggestion.

line 159: Is an optimisation type problem like the next paragraph being considered? If so, please say so, and if not, would suggest not calling it the "cost function" (e.g. "measure of skill", "skill metric" or similar)

To make this clearer, we rewrote lines 219-222 as 'For the hydrodynamic module, the vertical eddy viscosity component acting on the tidal flow $A_{v,ti}$ is calibrated. An optimisation procedure is employed for this variable to find the minimum error between the modelled and the observed M2 water level variations, using the skill score as used by Davies and Jones (1996), which computes a cost function f as'

line 170-171: "A score of NSE = 1 indicates perfect agreement, while NSE = 0 means that the model..."

We followed this suggestion.

line 171: NSE < 0 seems possible (small denominator and non-zero nominator), what would that mean?

We added after line 235: ' and finally *NSE* < 0 indicates that the observed mean is a better predictor than the model.'

line 174-175: Here the "b"s are in between LaTeX dollar signs I think, while the one at line 177 doesn't, inconsistency. I think it should not have the dollar signs? (Here I usually suggest the author to fix it, because who knows what the typesetter ends up doing.)

Indeed, the "b"s should be normal case letters. This is corrected in the revised version.

line 196: "...which reduces horizontal salinity gradients", "...which smooths out the salinity distribution", or similar

We followed this suggestion.

line 202: "...zero, while for the Waal..."

We followed this suggestion.

line 207: "...transport, while further inland..."

We followed this suggestion.

line 209: Here I would actually suggest bringing the equation and even some of the material in Appendix B up. Again, I get that the flow could be disrupted by excessive details, but the lack of key results even as a summary in the vicinity means the reader has to jump all the way to the appendix, and is unnecessarily abrupt.

We presented the equation for u'st in the revised version earlier in the text, following an earlier comment, and will refer in the revised version to this equation.

line 214: "...component due to tides $T_T$ is small..."

We followed this suggestion.

line 222: "...$T_T$ is a subdominant component..." (or "subdominant" -> "secondary")

We followed this suggestion.

line 225: "Around junctions such as..."

We followed this suggestion.

line 230, 233: Suggest "Panel a" to "Fig 6a" and similar for "b"

We followed this suggestion.

line 240: Quite a lot of acronyms going on, if reduction is possible that would be greatly appreciated...(this is a general point throughout the paper)

We agree that the use of the large number of acronyms is not ideal. However, we want to describe the characteristics of a number of channels and it is therefore inevitable that we use their acronyms. We avoid using acronyms in the titles of sections and in the conclusions.

line 294: "...factor of six smaller..."

We followed this suggestion.

line 316: "...simulations that have the..."

We followed this suggestion.

line 334: "...X_2 rather than a decrease."

We followed this suggestion.

line 335: "To understand why the scalings are different in a network, a simple model for the RW is constructed..."

We followed this suggestion.

line 337: Here I would actually suggest bringing the material in Appendix E up, certainly the key results, but possibly even the figure and/or the formulation.

In lines 413-425 it is described what is done in Appendix E, including the key results, and the resulting figure is in Fig 8c. We think this should be sufficient to inform the readers here.

paragraph starting line 348: Would strongly suggest removing the use of converse text in brackets here, mostly because I think there are better ways of achieving the same thing. I would suggest getting rid of all the text converse text in the brackets, and say somewhere that the converse case holds true (probably either at the beginning or at the end).

Suggestion will be followed. We removed the text in brackets and mention now after line 438 that the reverse holds for the shoaling.

line 421-422: would lessen the strength, "...a shallower Oude Maas leads to a decrease..." rather than "causes", because this is true in the model but there is no evidence to currently suggest it extends beyond that.

We followed this suggestion.